# Vertical distribution of aerosols over the Maritime Continent during El Nino

Jason Blake Cohen[1], Daniel Hui Loong Ng[2], Alan Wei Lun Lim[3], Xin Rong Chua[4]

[1]School of Atmospheric Sciences, Sun Yat-Sen University, Guangzhou, China
[2]Tropical Marine Science Institute, National University of Singapore, Singapore
[3]The Chinese University of Hong Kong, Hong Kong, China
[4]Princeton University, Princeton, NJ, USA

*Correspondence to*: Jason Blake Cohen (jasonbc@alum.mit.edu)

**Abstract.**

The vertical distribution of aerosols over Southeast Asia, a critical factor impacting aerosol lifetime, radiative forcing, and precipitation, is examined for the 2006 post El-Nino fire burning season. Combining these measurements with remotely sensed land, fire, and meteorological measurements, and fire plume modeling, we have reconfirmed that fire radiative power is underestimated over Southeast Asia by MODIS measurements. These results are derived using a significantly different approach. The horizontally constrained Maritime Continent's fire plume median height, using the maximum variance of satellite observed Aerosol Optical Depth as the spatial and temporal constraint, is found to be $2.04 \pm 1.52$km during the entirety of the 2006 El Nino fire-season, and $2.19 \pm 1.50$km for October 2006. This is 0.83km (0.98km) higher than random sampling and all other past studies. Additionally, it is determined that 61(+6-10)% of the bottom of the smoke plume and 83(+8-11)% of the median of the smoke plume is in the free troposphere during the October maximum; while correspondingly 49(+7-9)% and 75(+12-12)% of the total aerosol plume and the median of the aerosol plume, are found in the free troposphere during the entire fire-season. The vastly different vertical distribution will have impacts on aerosol lifetime and dispersal. Application of a simple plume rise model using measurements of fire properties underestimates the median plume height by 0.26km over the entire fire season and 0.34km over the Maximum fire period. It is noted that the model underestimation over the bottom portions of the plume are much larger. The center of the plume can be reproduced when fire radiative power is increased by 20% (with other parts of the plume ranging from an increase of 0% to 60% depending on the portion of the plume and the length of the fire season considered). However, to reduce the biases found, improvements including fire properties under cloudy conditions, representation of small scale convection, and inclusion of aerosol direct and semi-direct effects is required.

# 1. Introduction

Properly quantifying the vertical distribution of aerosols is essential to constrain their atmospheric distribution, which in turn impacts the atmospheric energy budget [*Ming et al.*, 2010; *Kim et al.*, 2008], circulation, clouds and precipitation [*Tao et al.*, 2012; *Wang* 2013], and human health [*Burnett et al*, 2014]. However, there are complicating factors including spatial and temporal heterogeneity in emissions [*Cohen and Wang*, 2014; *Cohen*, 2014; *Giglio et al.*, 2006; *Petrenko et al.*, 2012; *Wooster et al.*, 2012], and uncertainties and non-linearities associated with aerosol processing and removal from the atmosphere [Tao et al., 2012; *Cohen* and *Prinn*, 2011; *Cohen et al.*, 2011]. Furthermore, a lack of sufficiently dense measurements leads to difficulty constraining the measured distribution of aerosols over scales from hundreds to thousands of kilometers or over time frames on the decadal to longer time scales [Cohen and Wang, 2014; *Delene* and *Ogren*, 2002; *Dubovik et al.*, 2000; *Cohen* et al., 2017].

Models are very poor at reproducing the actual vertical distribution of atmospheric aerosols [*Cheng et al.*, 2012; *Schuster et al.*, 2005; *Tsigaridis et al.*, 2014]. They also tend to strongly underestimate the total atmospheric column loading of aerosols [*Colarco et al.*, 2004; *Leung et al.*, 2007]. Furthermore, vertical measurements are sparse, and in many regions do not provide adequate statistics to make informed comparisons with real world conditions. This is no more apparent than over Southeast Asia, where model studies [*Tosca et al.*, 2011; *Martin et al.*, 2012] have concluded that almost all aerosols are narrowly confined in the planetary boundary layer, although measurements demonstrate otherwise [*Lin et al.*, 2014]. Presently, there are no known modeling efforts that have been able to reproduce this significant atmospheric loading and the ensuing vertical distribution.

Additionally, aerosol emissions databases in Southeast Asia are quantified using a bottom-up approach, where small samples and statistics of the activity, land-use, economics, population, and hotspots are aggregated [*van der Werf*, 2010; *Lamarque*, 2010; *Bond et al.*, 2004]. This problem is further exacerbated by the fact that emissions from organic soils are already not well studied even in non-tropical regions (Urbanski, 2014). This generally leads to sizable bias: there are few measurements and rapidly changing land-surface features. A recent couple of papers has used measurements and models in tandem to quantify a significant underestimation in aerosol emissions over Southeast Asia. This underestimation occurs both in terms of magnitude [*Cohen and Wang*, 2014] as well as the spatial and temporal distribution of the emissions [*Cohen*, 2014]. In specific, it is significantly impacted on an interannual and intraannual basis by fires.

The vertical distribution is further ill-constrained due to an incomplete understanding of in-situ production and removal mechanisms which are dependent on washout, which itself is also poorly modeled [*Tao et al.*, 2012; *Wang* 2013], especially in the tropics during the dry season [*Petersen and Rutledge*, 2001; *Ekman et al.*, 2012]. Heterogeneous aerosol processing may also change the hygroscopicity, which in turn impacts the washout rate and vertical distribution of the aerosols [*Kim et al.*, 2008; *Cohen et al.*, 2011]. These factors have been shown to combine such that small changes in the initial vertical distribution can lead to differences in atmospheric transport thousands of kilometers apart [*Wang,* 2013].

The Maritime Continent of Southeast Asia has faced widespread and ubiquitous fires the past few
decades, due to expanding agriculture, urban development, economic growth, and changes in the base
climatology that induce drought [*Center*, 2005; *Dennis et al.*, 2005; *van der Werf et al.*, 2008; *Taylor*,
2010]. These fires contribute the major fraction of the atmospheric aerosol burden during the dry season
[*Cohen*, 2014]. However, these fires are unique: they are relatively low in radiative power and temperature
yet cover a massive net surface area, making their statistics and extent hard to characterize from remote
sensing. Their total emissions are very high, and thus during the burning season they dominate the aerosol
optical depth (AOD) and $PM_{2.5}$ levels over thousands of kilometers [*Field et al.*, 2009; *Nakajima et al.*,
1999]. Due to their widespread and dispersed nature, the fires as a whole in this region are geospatially
coherent in timing and geography, although they may individually burn for different lengths of time, as a
function of localized precipitation and soil moisture, and global circulation patterns such as El-Nino
[*Cohen*, 2014; *Wooster et al.*, 2012; *Hansen*, 2008].
A comprehensive previous attempt to study aerosol height over Southeast Asia was performed by
Lee et al. [2016]. They used the total The Cloud-Aerosol Lidar with Orthogonal Polarization (CALIOP)
profile, but were not specific about how they cleared or accounted for high ice clouds that frequently found
in this part of the world. They also used day-time data without considering the issues of solar reflection and
backscatter [Winker et al., 2013]. Furthermore, they used satellite derived single scattering albedo (SSA)
approximated by each pass, although this product has been shown to be highly error-prone over Southeast
Asia [Rogers et al., 2009; Hostetler, 2008]. This work did not address how the spatially-disparate individual
path measurements from CALIOP were analyzed or separated in terms of those sampling parts of the fire
plume as compared to those sampling regions not impacted by fires, such as in Cohen [2014] and Cohen et
al. [2017]. Over this region of the world, there has been no direct local validation of the CALIOP product
by other LIDAR related instruments [Sugimoto et al., 2014a]. The only comparisons made so far have been
model-based validation studies [Campbell et al., 2013].
This work describes a new approach to comprehensively sample the vertical distribution of smoke
aerosols, by first using decadal scale measurements of AOD from the Multi-angle Imaging
SpectroRadiometer [MISR] satellite [*Cohen*, 2014], and then separating the smoke impacted regions by
the magnitude of the measured variability. During the 2006 El-Nino enhanced burning, one of the 2 largest
such events over the past 15-year measurement record, this approach yields a much higher vertical aerosol
height than the traditional random sampling approach. A simple plume-rise model [*Achtemeier et al.*, 2011;
*Briggs*, 1965] using reanalysis meteorology [*Kalnay et al.*, 1996] and measured fire properties was found to
underestimate the measured heights. However, the model could be improved to match the median heights
by increasing the measured fire radiative power [*Sessions et al.*, 2011; *Sofiev et al.*, 2012]. This finding
implies that measured fires may be underestimated in terms of their strength, or that there are missing fires.
However even with scaling, the top and bottom heights of the measured plume still cannot be reproduced.
The data shows that an improved representation of both localized convective transport and the aerosol
direct and semi-direct effects [*Ekman et al.*, 2007; *Wang*, 2007] are required to make further improvements.
It is hoped that these results will provide insight to those working on understanding the strong 2015-2016
El-Nino conditions.

## 2. Methods

### 2.1 Geography

This work is focused on the Maritime Continent, a sub region of Southeast Asia ($8^o$S to $8^o$N, $95^o$E
to $125^o$E) (**Figure 1**) that experiences wide-spread and highly emitting fires on a yearly basis during the
local dry season (starting in August/September and proceeding continuously through October/November).
The combined magnitudes of so many small fires effectively produces a single massive smoke plume in the
atmosphere, that covers much of the region, extending thousands of kilometers [*Cohen*, 2014]. These wide
spread fires are due to anthropogenic clearing of rainforest and agriculture [*Cohen* et al., 2017; *Dennis et
al.*, 2005; *van der Werf et al.*, 2008; *Taylor*, 2010; *Miettinen et al.*, 2013; *Langmann et al.*, 2009]. Over this
region, during the dry season, the removal of aerosols is quite slow, leading to the overall properties of the
plume being relatively consistent over space and time [*Cohen*, 2014]. Therefore, the overall properties of
the smoke plume, when correctly bound in space and time, can be robustly related to the overall properties
of individual fires and daily measurements of AOD from the MISR satellite (**Figure 1**) [*Cohen*, 2014].
In 2006, the El-Nino conditions led to an enhanced drought, with subsequent fires lasting from
early September through mid-November. To ensure that this event is uniquely and completely analyzed,
data from September $3^{rd}$ through November $9^{th}$ is ultimately used (more details are given in **Figure 2 and
Figure 3a**, which are defined later). The region in (**Figure 1**) consists of the EOF larger than 2.2 (Bjornsson
and Venegas, 1997; Cohen et al., 2017) as calculated from the measured MISR AOD. This region forms the
boundary of the fire source regions (over land) and downwind regions (over both land and sea). This
approach analytically provides a holistic representation in space and time of the impact of individual fires
on the large-scale structure of the aerosol plume. Therefore, the approach allows the vertical distribution of
the smoke to be comprehensively sampled, including those obscured by clouds (very common in this
region), and aged aerosols which were emitted in the fire and transported significantly downwind.

### 2.2 Measurements

The CALIOP instrument is an active lidar, quantifying the vertically resolved atmospheric
backscatter strength at 532 nm and 1064 nm (a reasonable approximation of the vertical profile of
aerosols), and and polarization at 532 nm. The combination of these measurements allows an indication of
particle size (large or small) and whether the particle is a cloud or an aerosol [*Winker et al.*, 2003].
Specifically, we use the backscatter at 532nm and the vertical feature mask (vertical resolution 30m below
8.2km and 60m from 8.2km to 20.2km, horizontal resolution 1/3km) [*Hostetler et al.*, 2006]. To avoid
issues of cloud contamination and solar reflectance only night time data only is used, and any identified
clouds are removed [Winker et al., 2013].
Since the width of each pass is narrow, it is not spatially representative in general. However, given
the relative consistency of the plume as a whole, samples constrained within the plume's spatial extent,
taken on the same day, are statistically representative of the smoke plume as a whole [*Cohen*, 2014]. This
approach improves upon the approach of Winker et al. [2013] by relaxing the uniform "horizontal box
size". Instead, the area of analysis is constrained so that in a more general spatial and temporal domain
based on a homogeneous response in measurement space. Specifically, by constraining the region using
AOD, each region therefore has a much larger number of lidar measurements that are consistent with the
physical effects occurring within the region, thereby allowing for improved statistical representation.

The extinction-weighted top (10% vertically integrated height), middle-upper (30% vertically

integrated height), median (50% vertically integrated height), middle-lower (70% vertically integrated
height), and bottom (90% vertically integrated height) are computed for each individual measurement, with
the values retained if the aerosol is not in the stratosphere (assumed to be 15km) (**Supplemental Figure 1**).
The data is then aggregated first by day, and second by geography, either into the fire-impacted region, or
the non fire-impacted region, based on (**Figure 1**) [*Cohen*, 2014]. The aggregated set of measurements is
used to compute probability densities and statistics, demonstrating the vast difference over the fire-
impacted and non-fire impacted regions (**Figures 3a,3b)**. The vertical heights both significantly higher and
less variable ($p<0.01$) over the fire region than the non-fire region, inclusively from September 3[rd] through
November 9[th].

Measurements of aerosol optical depth (AOD) [*Kaufman et al.*, 2003], fire radiative power (FRP)

and fire temperature ($T_F$) [*Freeborn et al.*, 2014; *Ichoku et al.*, 2008] are obtained from the MODIS
instrument aboard both the TERRA and AQUA satellites. Version 5, level 2, swath-by-swath measurements,
at daily resolution are use for AOD (best solution 0.55 micron), with a spatial resolution of 10km by 10km,
and FRP/$T_F$, with a spatial resolution of 1km by 1km. Given the prevalence of clouds in this region, the
cloud-cleared products are used, leading to a possible low bias in the FRP/$T_F$ measurements, as well as
some fires not measured at all [*Cohen* et al., 2017; *Freeborn et al.*, 2014; *Ichoku et al.*, 2008; *Kahn et al.*,
2008; *Kahn et al.*, 2007]. On the other hand, while some grids are contaminated, the sheer spatial distance
of the plume and the fact that the overwhelming majority of atmospheric aerosols during this time of the
year are due to fires. In fact, there is no observable bias in the overall statistics of the measured AOD
[*Cohen*, 2014] as observed by looking at the spatially averaged MODIS AOD and statistics over the fire-
constrained and non fire-constrained regions (**Figure 2**). The AOD is higher ($p<0.01$) over the fire-
constrained region, from September 3[rd] through November 9[th], making the findings consistent with the
approach employing the 12-years worth of MISR measurements, as well as the results from the CALIOP
observations already discussed.

In terms of MODIS retrieval uncertainties over land, especially during fire events, there are two

important issues to consider. The first is that under extremely high AOD conditions (AOD>2), frequently
aerosols are flagged/reclassified as clouds, which brings about a negative bias. This bias would lead to an
even higher AOD over the fire plume region if it were properly handled, leading to an even larger
difference between "fire region" and the "non-fire region". The second is the error in the over-land retrieval
can go as high as 15%. However, based on the results in (**Figure 2 and Supplemental Figure 2**), the
difference between the "fire region" and the "non-fire region" is statistically sound even assuming the error
is larger than 15%. It is also the reason why MISR was used for the initial definition of the two regions,
since its ability to cloud clear is better than MODIS over this region [Kahn et al., 2010].

While there are many errors involved with using the satellite data, the errors in this case are

sufficiently small as to not impact the analysis and results over Southeast Asia during the fire season
(Cohen, 2014; Cohen et al., 2017). The AOD and certain surface products, when used to run models,
have been found to compare in magnitude, spatial, and temporal extent, to various ground based
surface and column measurements, such as from Aerosol Robotic Network [AERONET], the United
States National Oceanic and Atmospheric Administration surface measurement network [NOAA], and
other available air pollution networks. The data-driven models have been shown to lead to a reduction
in the annualized RMS error as compared with the Intergovernmental Panel on Climate Change
Representative Concentration Pathways [IPCC RCP] emissions scenarios by a factor of 2 to 8 against
AERONET stations throughout Asia (Cohen and Wang, 2014). Furthermore, on a month-to-month
basis, the results of the data-driven models have been shown to lead to a reduction in the RMS error
by a factor of 1.8 and of an improvement in the coefficient of determination statistic [$R^2$] by a value of
0.2 to 0.3, when compared against the Global Fire Emissions Database [GFED] dataset (Cohen 2014;
Cohen et al. 2017). Given these findings, it is reasonable to assume that the methodology is as reliable
as anything else presently available.

## 201    2.3 Plume Rise Model

A simple model is employed to simulate the height to which a parcel of air initially at the surface

over the fire will rise, based on buoyancy, vertical, and horizontal advection (**Supplement**). The
formulation requires information about the temperature and radiative power of the fire as well as local
meteorology [*Achtemeier et al.*, 2011; *Briggs*, 1965], and yields an idealized height to which aerosols
emitted will rise. The buoyant plume rise is a thermodynamic approximation in nature and thus not as
physically realistic as a large eddy approach, which solves the atmospheric fluid dynamical equations by
parameterizing turbulence at the scale of tens of meters. However, it is less computationally expensive and
more generalizable in the context of approximating the thousands of fires spread geographically over
hundreds of thousands of square kilometers. On the other hand, it is more physically realistic than empirical
relationships from multi-angle measurements [*Sofiev et al.*, 2012], which have also been attempted, but
show poor performance in Southeast Asia.

These relationships are efficiently solved using measurements of meteorological and fire

properties, allowing them to be used as rapid parameterizations within regional or global models. However,
there are errors associated with reconciling the different temporal and spatial scales of reanalysis
meteorology, especially convection and associated transport. Secondly, cloud-cover in this region leads to
both missing fires and low-bias in measurements of fire properties [Sofiev et al., 2012; *Kaufman et al.*,
2003]. Third, the cloud-cover also leads to a heavier contribution of model results in the reanalysis
meteorology. Finally, the effects of the optically thick aerosol plume's feedback on the radiative profile is
likely important, but beyond the scope of this work and hence not taken into consideration [*Ekman et al.*,
2007; *Wang*, 2007].

## 3. Results and Discussion
### 3.1 Measured Aerosol Vertical Distribution
The fire-constrained aggregated daily statistics of the measured vertical aerosol height from
CALIPSO [*Winker et al.*, 2003] is given in (**Figure 3a**), with the aggregated statistics from the October
fire-maximum time and (*the entirety of the fire season*) over the fire-constrained region of the bottom,
middle-lower, median, middle-upper, and top heights respectively: $1.68 \pm 1.55$km ($1.49 \pm 1.58$km),
$1.92 \pm 1.51$km ($1.76 \pm 1.54$km), $2.19 \pm 1.50$km ($2.04 \pm 1.52$km), $2.53 \pm 1.51$km ($2.38 \pm 1.54$km), and
$3.03 \pm 1.52$km ($2.91 \pm 1.57$km) (**Table 1**). These results are supported by the statistical values of aerosol
heights measured by the MPL station in Singapore throughout the period from September 1 to November
30, 2015 (**Supplemental Figure 3**), which are found to range from 1.6km to 2.4km. 2015 was selected to
compare against ground-based lidar measurements, since it was an El-Nino year, and there were no such
measurements available from 2006. It is also known that 2015 in Singapore contained large amounts of
aerosols advected to Singapore from downwind burning sources. Overall, the close resemblance between
these years allows inference from the results.

On the other hand, the non fire-constrained region's aggregated statistics of the measured vertical
aerosol height is quite different (**Figure 3b**), with the respective bottom, middle-lower, median, middle-
upper, and top heights during the October maximum-fire period being: $0.65 \pm 0.98$km, $0.93 \pm 0.98$km,
$1.21 \pm 1.00$km, $1.53 \pm 1.02$km, and $1.98 \pm 1.08$km (**Table 1**). The average aerosol height over the fire-
constrained region is both much higher and more variable at every vertical level as compared to the non
fire-constrained domain. This difference leads to 61(+6-10)% of the bottom of the smoke plume and 83(+8-
11)% of the median of the smoke plume in the free troposphere during the October maximum; while 49(+7-
9)% and 75(+12-12)% of the respective bottom and median of the aerosol loading is in the free troposphere
over the entirety of the fire-season, over fire-constrained domain. On the other hand, only 17(+10-9)% of
the median of the aerosol loading is located in the free troposphere over the non fire-constrained domain
during the October maximum fire period. However, the variability is roughly constant at all levels over the
fire-constrained region, while the variability increases with vertical level, over the non fire-constrained
region. These results are based on more than 10,000 daily CALIOP measurements.

All three findings, higher average aerosol height, larger variance of height, and a consistent
variance of height at all levels, are consistent with areas where most of the aerosol loading is due to surface
fires. Firstly, the buoyancy from fires increases the expected height, with differences in buoyancy from
different strength fires producing random variability in the measured heights. So long as the distribution of
fire strength and meteorology do not differ too much from day-to-day, the variance in aerosol heights
should also not vary much. On the other hand, over non fire-constrained regions, the major contribution to
the vertical aerosol variability is convection, which is expected to increase in variability the higher one
moves upwards from the surface.

Furthermore, the relatively constant variability across the heights in the fire-constrained region is
consistent with a proposed radiative-stabilization effect. The extremely high measured AOD values found
by MODIS [*Kaufman et al.*, 2003] over the fire-constrained domain (from 0.5 to 2.0, with most days over
1.0), leads to observable surface cooling (**Figure 2**). Additionally, black carbon aerosols [BC] emitted from
the fire, absorbs incoming solar radiation near the upper portion of the plume, providing a source of
warming. This combination leads to additional stabilization of the atmosphere, and therefore reinforces the
observed vertical aerosol distribution.
These results are thus consistent with the observed reduction in in-situ vertical processing over the
regions downwind from the fire sources, but still within the fire-constrained plume region, where buoyancy
from the fires and the self-stabilization effect seem to contribute more than random deep convection.
However, over the non fire-constrained region, given the low AOD and lack of fires, both of these effects
are not observed, and convection dominates, which is consistent with the less uniform vertical distribution.
Given these clear and observed differences, only results from the fire-constrained region will be considered
further.
A significant amount of aerosol mass exists in the free troposphere over this region. Assuming the
measured boundary layer height can be represented by the range from 700m to 1300m, with a central value
of 1000m (as observed in Singapore [*Chew et al.*, 2013]) and applied over the domain, the resulting total
loading of aerosols over the boundary layer can be computed. This value, when applied over the entire
geographical domain, the amount of measurements above the boundary layer in October is found to be
[67,61,51]%, [80,70,61]%, [91,83,72]%, [96,92,83]%, and [99,97,94]% respectively of the bottom, lower-
middle, median, upper-middle and top extinction. Although October is slightly more intense, the same
pattern, just to a slightly lesser extent, is found throughout the entire season, with [56,49,40]%,
[72,61,51]%, [87,75,63]%, [96,90,77]%, and [99,97,93]% of the measurements respectively of the bottom,
lower-middle, median, upper-middle and top extinction. This is much higher than previous studies, which
indicated most of the smoke remained within the boundary layer [*Tosca et al.*, 2011].
Analysis of the daily measured heights demonstrates 3 statistically unique days: October 11[th], 15[th]
and 22[nd] (**Table 2**). On the 11[th], the top and upper-middle measurements fall within the top 15%, while the
median measurements fall within the top 20% of the month's measurements, implying that the result is
consistent with a deep, single layer, extending throughout the lower and middle free-troposphere. The 15[th]
and 22[nd], while not being as high in the middle-troposphere, also have little to no aerosol in the planetary
boundary layer due to being more confined in the vertical, implying a narrow layer in the middle free-
troposphere. These results are consistent with the measured aerosol layer being mostly in the free
troposphere, a result that is not consistent with the measured FRP or meteorology. This leads to two
important implications. First, that aerosol lifetime on these days will be considerably longer than models
typically reproduce, and thus the radiative forcing will be considerably more warming. Secondly, that the
typical modeling approach which places fresh aerosols directly emitted from the surface, to the given top of
the plume, is likely not true. These are two serious issues impacting the ability of most models to be able to
correctly capture the aerosol loading.

On the remaining days, the measured heights are consistent on a daily average basis with relatively

uniform emissions, meteorology, and vertical buoyant rise. Although there is some intense but
heterogeneous forcing impacting the vertical distribution, such as localized convection and aerosol cloud
interactions, these are generally not observed to bias the overall plume's properties. Only on October 11[th],
15[th], and 22[nd], are there higher heights or a narrower vertical structure, combined with no readily available
explanation to be found in the fire, AOD, or meteorological properties on these days. This combination can
only be explained by either a clear change in the convection on those days, or some other phenomena not
considered in or otherwise represented by the reanalysis meteorology. The robustness of this approach
assures the validity of these results over the region and time period herein.

A comparison between the inverse model by Campbell et al. [2013; Supplemental Figure 6] and

this work's underlying Kalman Filter plus variance maximization modeled fields, shows that this new
modeling approach performs better during the biomass burning season [Cohen, 2014; Cohen and Wang,
2014; Cohen et al., 2017]. Furthermore, the results found using the approach employed here, match well
with individual measurement campaigns lead by Lin Neng-Hui, et al. [2013, 2014, etc.], and the AD-Net
measurement network [Sugimoto et al, 2014b]. The common finding is a small number of on-the-ground
lidar at multiple places within the Northern portion of Southeast Asia and Greater East Asia also observe
something similar. However, since the geographic regions are not identical, therefore they cannot be used to
directly validate the region studied here. But, there is a sufficient amount of similarity, to make an
anecdotal connection. Given these factors, we present the results here as the best available for use at this
time, when targeting this region of the world during the biomass burning season.
## 3.2 Measured Fire and Meteorological Properties

The daily aggregated measurements of fire radiative power (FRP) [*Freeborn et al.*, 2014; *Ichoku et*

*al.*, 2008] indicate there are 109395 actively burning 1kmx1km pixels in October 2006. However, filtering
for high confidence [Level 9] active fires, reduces this number to 6941 1kmx1km pixels. The respective
measurements have 10%, median, and 90% values of FRP of [115,300,975] W/m$^2$ for all fires and
[185,540,1495] W/m$^2$ for high confidence fires (**Table 3**). Overall, these values are much lower than FRP
measured over other intensely burning regions [*Giglio et al.*, 2006]. However, the results are consistent
with the fact that fires in the Maritime Continent occur under relatively wet surface conditions, due to high
levels of mineral-soil moisture, extensive peat, and intermittent localized precipitation [*Couwenberg et al.*,
2010]. These results are based on more than 3000 daily MODIS fire hotspots and associated meteorological
measurements.

There is only one day, October 2[nd], with a statistically high FRP (daily mean more than monthly

90% value), for high confidence fires. Similarly, there are two days, October 28[th] and 30[th], with an
abnormally low FRP (daily mean less than monthly 15% value), for high confidence fires. None of these
days have a statistically abnormal fire vertical height distribution. However, October 28[th] and 30[th] both

330 show a sizable increase in AOD over the fire constrained region, with the AOD more than 2 standard

331 deviations greater than the mean over the non fire constrained region, as compared to the period of time

332 from the 25th through the 27th. One consistent rationale is that there was large-scale precipitation event at

333 that time, which in turn both increased aerosol removal and wetting of the surface. This in turn led to lower

334 temperature and FRP and correspondingly higher aerosol emissions factor on these days. Overall, there is

335 no apparent impact of day-to-day variability of measured FRP driving observed variation in measured

336 aerosol heights, and hence only high confidence fire data is subsequently used.

337  To examine this hypothesis, the GPCP [Global Precipitation Climatology Project] One-Degree

338 Daily Precipitation Data Set of global precipitation has been employed to study the amount and duration of

339 rainfall over the fire-burning and non fire-burning regions [Huffman et al., 2012]. A spatial/temporal

340 analysis of this dataset, over both the Fire Region and the No-Fire region confirms this hypothesis

341 (**Supplemental Figure 4**). Overall, there was considerably lower rainfall over the Fire Region than the No-

342 Fire Region, however, on all days that there was a decrease in AOD and FRP over the Fire Region, there

343 was a heavy Rainfall at the same time, or one or two days before. The measurements have a correlation

344 coefficient of -0.39 with a corresponding $p<0.01$. There is no other statistically significant correlation found

345 over any other combination of the regions with any other combination of rainfall.

346  The Modern-Era Retrospective Analysis for Research and Applications [MERRA] [Rienecker et

347 al., 2011] reanalysis meteorology is used for the horizontal and vertical wind, and vertical temperature

348 profile at each location where a fire is measured (**Table 3**). MERRA was chosen because it is based on

349 NASA satellite measurements, and thus should be more consistent with the measurements used here. With

350 the exceptions of October 5[th] and 20[th], the horizontal wind is relatively calm $6.0 \pm 1.3$m/s. Also, throughout

351 the entire month, the vertical temperature gradient is relatively stable $-5.45 \pm 0.16$K/km, with only 7

352 individual fires occurring under unstable atmospheric conditions. Therefore, dynamical instability is not

353 expected to contribute greatly to the vertical distribution [*Stone and Carlson*, 1979]. Also, the role played

354 by the large-scale vertical wind is small $2.1 \pm 1.6$mm/s. Given the atmospheric stability and fire-controlled

355 buoyancy conditions, the plume rise model approach should offer a reasonable approximation of the aerosol

356 vertical distribution.

357  The approach used here relies upon the atmosphere being either stable or only barely non-stable.

358 In this part of the world there are two reasons that contribute to most fires occurring under such conditions.

359 Firstly, that major instability frequently leads to rain, fire suppression, and aerosol wash-out. Secondly that

360 induced surface cooling and atmospheric heating by the extensive aerosol layer itself tends to increase

361 atmospheric stability. Such points are made clear in terms of the major unaccounted for processes in the

362 MERRA data at this resolution: localized convection (due to model resolution), and aerosol cooling and in-

363 situ heating effects (not incorporated into the underlying model). In theory the direct and semi-direct effect

364 may be able to be parameterized, but this would require a higher order model. Since these conditions and

365 effects are not considered by the plume rise model, they therefore cannot be explanations for discrepancies

366 in the modeled vertical distribution.

## 3.3 Modeled Aerosol Vertical Distribution

Applying the plume rise model, the aggregated daily statistics of the vertical aerosol height at the bottom, lower-middle, median, upper-middle, and top for the October fire-maximum time and (*the entirety of the fire season*) are 0.60km (0.41km), 1.14km (0.88km), 1.85km (1.40km), 2.87km (2.25km), and 4.99km (3.95km) respectively (**Figure 4, Table 4**). The mean of the daily median, lower-middle, and bottom modeled heights are consistently lower than the respective mean of the measured heights for the October fire-maximum time and (*the entirety of the fire season*) by 0.34km (0.64km), 0.78km (0.88km), and 1.08km (1.08km) respectively. The day-to-day differences show that the model generally underestimates the measurements, with the minimum and maximum differences between the two both ranging from -0.92 km to 1.36km, -0.63 km to 2.20km, and -0.19 km to 3.02km, respectively. The upper-middle modeled height is about equal to measurements, with a mean difference for the October fire-maximum time and (*the entirety of the fire season*) of an underestimate of 0.34km over the October maximum to an overestimate of (0.13km) through the entire fire season. The associated day-to-day variations are wide, but are roughly centered around zero, and vary from -1.22km to 1.06km. Finally, the top modeled heights are considerably higher than measurements, with an average overestimate for the October fire-maximum time and (*the entirety of the fire season*) being 1.96km and (1.04km) respectively. The day-to-day difference between the model and the measurements generally overestimates the measurements, with a value varying from -1.54 to 0.81km.

The model underestimates the height of the median through bottom of the plume, while simultaneously overestimating the top. First, this means that the model is not accounting for enough energy to obtain the average rise of the plume. At the same time, the modeled vertical spread is too large, implying other factors limit the height gain near the top of the plume and enhance the height near the bottom. The results are consistent with one or both of the two hypothesized effects; first, that a low bias exists in the measured values of FRP [*Kahn et al.*, 2007; *Kahn et al.*, 2008], leading to insufficient buoyancy. Second, that in-situ stabilization occurs due to aerosol radiative cooling in the lower parts of the plume and aerosol radiative heating within the upper parts of the plume. This combination of factors is also consistent with the observed underestimate in measured FRP to match the median height, as well as the hypothesized complete non-detection of small fires [*Kaufman et al.*, 2003]. There are also uncertainties in the MERRA reanalysis products, but given the large sample size and the narrowness of the MERRA distribution, the impact of these uncertainties is around 10%, which as we show later is considerably smaller than changes in the FRP.

A sensitivity analysis is used to quantify the effects of a low bias in FRP, by applying a constant multiplicative factor to the measured FRP for each fire, from 1.0 to 2.0 in steps of 0.1 (although only the results in steps of 0.2 are given in **Table 4**). Although there are also uncertainties associated with measured vertical wind and temperature structure, this is not considered (**Table 3**), since there is no way to couple meteorological effects at sub-grid scale, or otherwise not included in the reanalysis meteorology. The results are obtained by minimizing the root-mean square (RMS) difference between the daily measured and modeled heights, for each FRP scaling factor, at each of the middle-upper, median, and middle-lower levels. The respective best-fit enhancement factors over the October fire maximum (and the entire fire

season) are **1.0 (1.0)** for middle-upper measurements, having an RMS error of 0.69km (0.66km); **1.2 (1.2)**
for median measurements, having an RMS error of 0.78km (0.72km); and **1.6 (1.4)** for middle-lower
measurements, having an RMS error of 0.92km (0.82km) (**Figure 4**).
Another source of uncertainty is due to the height of the boundary layer itself, which is also
uncertain, due to both a lack of measurements, and a poor ability of reanalysis and other global scale
products to simulate the boundary layer in this part of the world. As before, the model was run in a
sensitivity mode, assuming 3 different average boundary layer heights. The results for the middle-upper,
median, and middle-lower levels best fit values over the October fire maximum (and the entirety of the fire
season) are enhancements of 1.0, 1.4, and 1.8 and (1.0, 1.1, and 1.5) respectively for a boundary layer
height of 1300m and 1.0, 1.3, and 1.6 and (1.0, 1.1, and 1.4) for a boundary layer height of 700m. These
results show that this factor is highly important in terms of modulating the magnitude of the best-fitting
FRP scaling factor. However, a similar biases still exists, where the model is reasonably good at
reproducing the upper-middle levels of the plume, but is incapable of reproducing the median and middle-
lower levels of the plume. Additionally, the larger values of the RMS error at the two more extreme
boundary layer heights lend further support to the initial supposition: overall the boundary layer height
throughout the fire region, lies within these boundaries.
Although there is no single best-fit FRP scaling factor, a reasonable fit of the model, based on
measured values from the middle-lower to the middle-upper plume levels can be obtained by using an
appropriate FRP enhancement. The results establish that current plume rise models can reproduce the
median vertical plume height over Southeast Asia by increasing the FRP by 20%, a finding consistent with
FRP generally underestimated over this region. By changing the FRP enhancement from 0% to 60%, the
central 40% of the aerosol plume's vertical extent can be modeled, although the top and bottom heights of
the plume cannot be reproduced. Additionally, the modeled plume is widely spread as compared to the
narrowness of the measured plume. Unfortunately, rectifying these limitations will likely require the use of
a more complex modeling approach and improvement of measured fire data.
There are additional errors associated with the non-complete complexity of the models employed.
The models do not capture the contribution of atmospheric stabilization due to both the direct and semi-
direct aerosol effects. Furthermore, these models do not take into account the impacts of localized
convection. However, the majority of other works that employ regional and global models use this exact
same methodology, and hence they also neglect these same small-scale phenomena in terms of
communication between the chemistry, radiation, and the meteorology.

## 4. Conclusions


This work quantifies the significant present-day underestimation of the vertical distribution of
aerosols over the Maritime Continent during an El-Nino influenced fire season, by introducing a new
method to appropriately constrain the measurements over the geographical region of the aerosol plume.
While this was a large-scale fire event, it was very special, because it occurred throughout almost all of
September, and all the way through the first third of November. Typically the wet-season arrives in this part
of the world sometime by the middle of October. And because of this, the wetness of the soil and the large-
scale meteorological flow, were both different this year from a more typical year. As a result, the measured
heights over the constrained region are found to be higher than previously thought. This year about 61(+6-
10)% of the bottom of the aerosol layer and 83(+8-11)% of the median of the aerosol layer being in the free
troposphere during the October maximum; while correspondingly 49(+7-9)% and 75(+12-12)% of the total
aerosol height and the median of the aerosol plume are found in the free troposphere during the entirety of
the fire-season. Due to the considerably higher vertical rise, the aerosols can be advected thousands of
kilometers from their sources and have a greater impact on the atmospheric and climatic systems.
Additionally, over the fire-constrained region, the vertical variability of the plume is found to be uniform
throughout its height, implying that it is controlled mostly by local forcing, such as the buoyancy released
by fires, localized convection, and aerosol/radiative feedbacks, such as the direct and semi-direct effects.

Application of a plume-rise model showed that there was an overall low bias against measured

heights. This is consistent with the FRP being underestimated in this region of the world due to large-scale
cloud cover. It was also determined that measured vertical heights are more narrowly confined in the
vertical than those simulated by models. A robust sensitivity analysis found that the middle-lower through
middle-upper extent of the plume can be reproduced if an appropriate (although changing) enhancement is
applied to the FRP ranging from 1.0*FRP to 1.6*FRP over the maximum period of the fire season, through
the month of October (and from 1.0*FRP to 1.4*FRP over the fire season as a whole, for most of
September, all of October, and the first third of November). Hence, the variable FRP enhancement factor
approach can allow for improved modeling of the height statistics for the middle-upper to middle-lower
extent of the plume.

However, it is not possible to reproduce either the top or bottom of the measured heights, the

knowledge of which is important to constrain the impacts of long-range transport and aerosol-climate
interactions. Nor is it possible to reproduce the narrow spread of the measured heights. The results are
consistent with the general understanding of current model shortcomings. Hence both the underestimation
of FRP values and current shortcomings in models need to be addressed, if we are to successfully model the
vertical aerosol distribution over this region of the world.

The results have been found to be robust over a region that behaves roughly uniformly over

thousands of kilometers, including regions both near and far from the source of the fires. Since there are
only a few days that have relatively unique aerosol and meteorological properties over the period studied,
the results support a few robust conclusions. First, if we want to improve the ability to model aerosol
heights, newer modelling approaches and improvements that will be able to resolve local-scale forcing,
such as deep convection, aerosol/radiation interactions, and aerosol-cloud interactions need to be
considered. Second, the biased underestimation of FRP is also an important point to improve the aerosol
height modeling, especially under conditions where cloudiness occurs or the measured AOD levels are very
high. These errors are exacerbated over regions where large-scale precipitation is very low or where there is
substantial aerosol/cloud intermixing. In all cases, until these model and measurement improvements are
made, there is expected to be a significant underestimation of the aerosol loadings and radiative forcing
distribution regionally, and to some extent globally. It is hoped that in the interim, the community will adapt
a variable enhancement of FRP in tandem with measurement-constrained boundaries of smoke plumes, as a
way to more precisely reproduce the statistics of the vertical aerosol distribution.
**Acknowledgements:**
We would like to acknowledge the PIs of the NASA MODIS, MISR, and CALIPSO projects for providing
the data. The work was supported by the Chinese National Young Thousand Talents Program (Project
74110-41180002), the Chinese National Natural Science Foundation (Project 74110-41030028), and the
Guangdong Provincial Young Talent Support Fund (Project 74110-42150003).

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

**Table 1**: Statistical summary of measured CALIPSO smoke plume heights in the El-Nino Season of 2006,
at different percentiles of extinction height (top/Z=10%, middle-upper/ Z=30%, median/Z=50%, middle-
lower/Z=70%, and bottom/Z=90%). The numbers in normal print correspond to the data during the
**maximum of the fire season** in **October**, while those numbers in *(italics)* correspond to the **entire fire**
**season** from **September 3rd through November 9th.** All data is further divided into the subset of the
Maritime Continent **impacted by smoke** (**FIRE**), and **not impacted by smoke** (**NO-FIRE**) (**Figure 1).**
"MEAN" is the average, "STD" is the standard deviation, and percentages XX% are the corresponding
distribution's percentiles.

| | bottom [km] | middle-lower [km] | median [km] | middle-upper [km] | top [km] |
|---|---|---|---|---|---|
| **FIRE 5%** | 0.18 *(0.17)* | 0.35 *(0.35)* | 0.56 *(0.57)* | 0.85 *(0.77)* | 1.27 (1.14) |
| **FIRE 10%** | 0.25 *(0.22)* | 0.48 *(0.46)* | 0.74 *(0.68)* | 1.06 *(1.02)* | 1.50 (1.47) |
| **FIRE 15%** | 0.30 *(0.26)* | 0.58 *(0.52)* | 0.88 *(0.77)* | 1.24 *(1.13)* | 1.64 *(1.60)* |
| **FIRE 50%** | 1.35 *(0.98)* | 1.58 *(1.33)* | 1.81 *(1.61)* | 2.18 *(2.00)* | 2.77 *(2.60)* |
| **FIRE 85%** | 2.73 *(2.59)* | 2.90 *(2.73)* | 3.11 *(2.91)* | 3.35 *(3.15)* | 3.70 *(3.67)* |
| **FIRE 90%** | 3.14 *(2.90)* | 3.29 *(3.13)* | 3.44 *(3.32)* | 3.66 *(3.57)* | 4.09 *(4.26)* |
| **FIRE 95%** | 4.19 *(4.25)* | 4.38 *(4.48)* | 4.70 *(5.08)* | 5.56 *(5.56)* | 5.65 *(6.02)* |
| **FIRE MEAN** | 1.68 *(1.49)* | 1.92 *(1.76)* | 2.19 *(2.04)* | 2.53 *(2.38)* | 2.91 *(3.03)* |
| **FIRE STD** | 1.58 *(1.55)* | 1.54 *(1.51)* | 1.52 *(1.50)* | 1.54 *(1.51)* | 1.57 *(1.52)* |
| **NO-FIRE 5%** | 0.16 | 0.33 | 0.48 | 0.60 | 0.70 |
| **NO-FIRE 10%** | 0.19 | 0.38 | 0.55 | 0.68 | 0.87 |
| **NO-FIRE 15%** | 0.21 | 0.42 | 0.59 | 0.77 | 1.12 |
| **NO-FIRE 50%** | 0.31 | 0.57 | 0.83 | 1.25 | 1.76 |
| **NO-FIRE 85%** | 1.16 | 1.64 | 2.01 | 2.36 | 2.85 |
| **NO-FIRE 90%** | 1.65 | 1.98 | 2.27 | 2.60 | 3.05 |
| **NO-FIRE 95%** | 2.22 | 2.45 | 2.73 | 2.99 | 3.41 |
| **NO-FIRE MEAN** | 0.97 | 0.98 | 1.00 | 1.02 | 1.08 |
| **NO-FIRE STD** | 0.65 | 0.93 | 1.21 | 1.53 | 1.98 |


**Table 2**: Summary of measured (CALIPSO) smoke plume heights over the entire fire season from
September 3rd to November 9th 2006, for days that are statistical outliers. The values here correspond to
having a mean value more than 85% of less than 15% **in bold**, or a mean value from 80% to 85% or from
15% to 20% in regular text. The levels are given as a percentile of extinction height over the subset of the
Maritime Continent impacted by smoke (fire-constrained), based on the MISR observations (**Figure 1).**

| | bottom (90% Extinction) [km] | middle-lower (70% Extinction) [km] | median (50% Extinction) [km] | middle-upper (30% Extinction) [km] | top (10% Extinction) [km] |
|---|---|---|---|---|---|
| **October 11th** | 2.29 | 2.54 | **3.26** | **4.11** | **4.93** |
| **October 15th** | 1.85 | 2.20 | | | |
| **October 22nd** | **2.55** | **2.85** | 2.95 | | |


**Table 3**: Statistics of measured fire properties (FRP and $T_F$), for all measured fires (**ALL**) and level 9
confidence fires (**L9**) and MERRA meteorological properties ($T_A$, v, U, dT/dz) corresponding to the
geographic locations of **L9**. All data is constrained by the boundaries of the fire extent, and is applicable to
results from the Maximum of the fire season corresponding to October 2006 (**Figure 1**). The distribution's
percentile is given as "**XX%**", the mean and standard deviation are given as "**MEAN**" and "**STD**". Note
that there were no observed fires of L9 on the following dates: 17[th], 22[nd], 23[rd], 24[th], 25[th], 26[th], 27[th], 29[th], 31[st].

|  | FRP **ALL** [W/m²] | FRP **L9** [W/m²] | $T_F$ **ALL** [K] | $T_F$ **L9** [K] | $T_A$ **L9** [K] | V **L9** [mm/s] | U **L9** [m/s] | dT/dz **L9** [K/km] |
|---|---|---|---|---|---|---|---|---|
| **5%** | 95. | 140. | 370. | 410. | 296.0 | 0.2 | 4.1 | -5.25 |
| **10%** | 115. | 185. | 390. | 445. | 296.4 | 0.4 | 4.4 | -5.27 |
| **15%** | 130. | 230. | 400. | 480. | 296.6 | 0.6 | 4.5 | -5.28 |
| **50%** | 300. | 540. | 535. | 725. | 298.4 | 1.5 | 6.0 | -5.43 |
| **85%** | 775. | 1240. | 910. | 1275. | 301.1 | 4.1 | 7.4 | -5.65 |
| **90%** | 975. | 1495. | 1070. | 1525. | 301.5 | 4.6 | 7.7 | -5.69 |
| **95%** | 1290. | 1855. | 1335. | 1850. | 302.1 | 5.6 | 8.1 | -5.75 |
| **Mean** | 510. | 920. | 702. | 1029. | 298.7 | 2.1 | 6.0 | -5.44 |
| **StD** | 720. | 1340. | 573. | 1057. | 2.0 | 1.6 | 1.3 | 0.16 |


**Table 4**: Statistics of the modeled fire heights corresponding to the maximum fire season of October and
the *(Entire fire season)*. All values are computed using level 9 confidence fires (**L9**) and MERRA
meteorology ($T_A$, v, U, dT/dz) at the corresponding geographic locations, with the daily average boundary
layer assumed to be 1000m. Sensitivity tests are shown with their respective weighting factor (**1.2, 1.4, 1.6,**
**1.8, or 2.0**) applied to the measured FRP. The modeled heights are given by percentile from the bottom
(5%) to the top (95%), while the mean and standard deviation are given as "**MEAN**" and "**STD**".  Note that
the model was not run on the following days, during which there were no observed **L9** fires: September
13[th],14[th],15[th],16[th],17[th],27[th], October 17[th], 22[nd], 23[rd], 24[th], 26[th], 27[th], and 31[st], and November 2[nd],9[th],14[th],16[th]
through 28[th],30[th].

|       | FRP(x1.0) [km] | FRP(x1.2) [km] | FRP(x1.4) [km] | FRP(x1.6) [km] | FRP(x1.8) [km] | FRP(x2) [km] |
|-------|----------------|----------------|----------------|----------------|----------------|--------------|
| **5%**  | **0.41** *(0.26)* | 0.44 *(0.30)* | 0.48 *(0.33)* | 0.53 *(0.35)* | 0.56 *(0.38)* | 0.60 *(0.41)* |
| **10%** | **0.60** *(0.41)* | 0.67 *(0.45)* | 0.73 *(0.49)* | 0.80 *(0.53)* | 0.85 *(0.57)* | 0.91 *(0.61)* |
| **15%** | **0.75** *(0.55)* | 0.83 *(0.61)* | 0.91 *(0.66)* | 0.98 *(0.72)* | 1.05 *(0.77)* | 1.12 *(0.82)* |
| **30%** | **1.14** *(0.88)* | 1.28 *(0.98)* | 1.40 *(1.07)* | 1.52 *(1.16)* | 1.63 *(1.25)* | 1.74 *(1.33)* |
| **50%** | **1.85** *(1.40)* | 2.07 *(1.58)* | 2.27 *(1.73)* | 2.47 *(1.88)* | 2.65 *(2.02)* | 2.82 *(2.15)* |
| **70%** | **2.87** *(2.25)* | 3.23 *(2.52)* | 3.54 *(2.76)* | 3.84 *(3.01)* | 4.12 *(3.23)* | 4.38 *(3.43)* |
| **85%** | **4.21** *(3.29)* | 4.66 *(3.67)* | 5.11 *(4.02)* | 5.53 *(4.35)* | 5.87 *(4.64)* | 6.22 *(4.92)* |
| **90%** | **4.99** *(3.95)* | 5.54 *(4.40)* | 6.08 *(4.80)* | 6.58 *(5.21)* | 6.97 *(5.56)* | 7.41 *(5.87)* |
| **95%** | **6.10** *(5.25)* | 6.79 *(5.86)* | 7.43 *(6.39)* | 7.76 *(6.83)* | 8.16 *(7.22)* | 8.61 *(7.57)* |
| **Mean** | **2.41** *(1.94)* | 2.69 *(2.17)* | 2.96 *(2.38)* | 3.21 *(2.58)* | 3.44 *(2.77)* | 3.67 *(2.95)* |
| **StD** | **1.98** *(1.76)* | 2.21 *(1.96)* | 2.42 *(2.15)* | 2.62 *(2.33)* | 2.81 *(2.50)* | 2.99 *(2.65)* |


**Figure 1:** Map of Maritime Continent. The smoke plume impacts the sub-region contained within the
dashed lines, or the so-called **fire-constrained** region. On the other hand, the region outside of the dashed
lines is the so-called **non fire-constrained** region. The colors on the plot correspond to the intensity of the
variance, as explained in Cohen [2014]. The plot is based on a variance maximization technique applied to
the measurements from all MISR overpasses from 2000 through 2014 (*Cohen*, 2014). Note that in this part
of the world 1 degree of latitude or longitude is approximately 100km, leading to a fire-impacted region
over 2500km across.

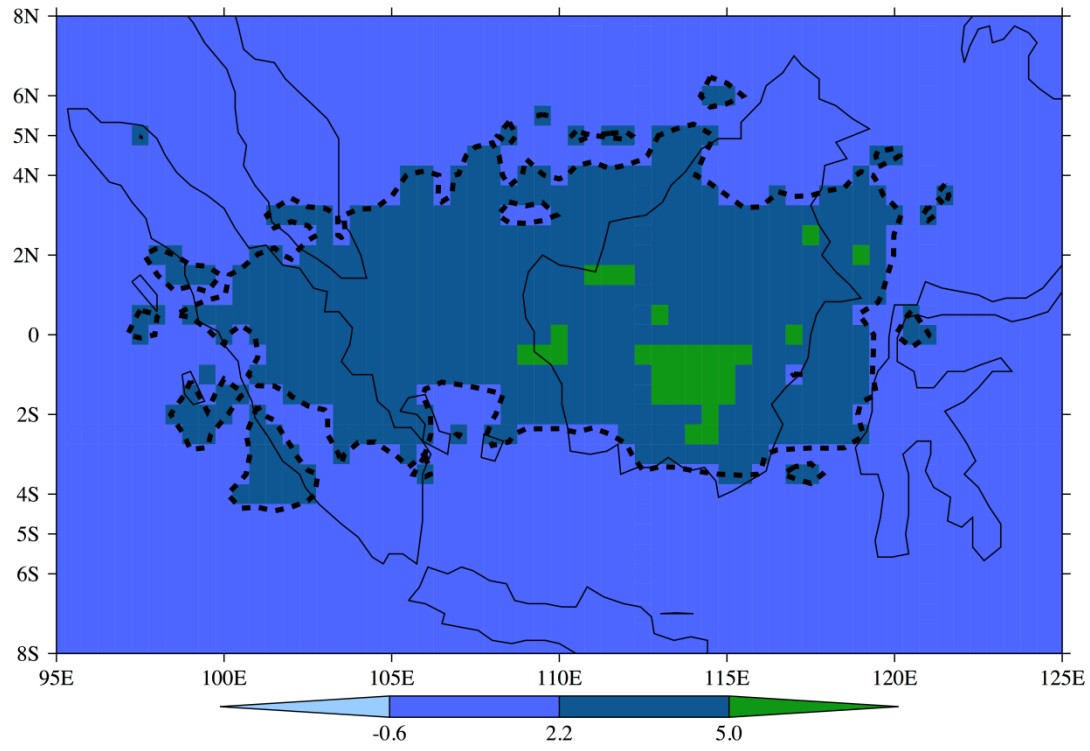


**Figure 2:** Time series of daily averaged measured AOD over the fire-constrained regions of the Maritime
Continent [blue], and the non fire-constrained regions of the Maritime Continent [red], as given in **Figure**
**1**. Circles are computed daily mean values, while dots are computed daily standard deviation bands. Note
that this figure contains the daily data from September 1, 2006 through November 30[th], 2006.

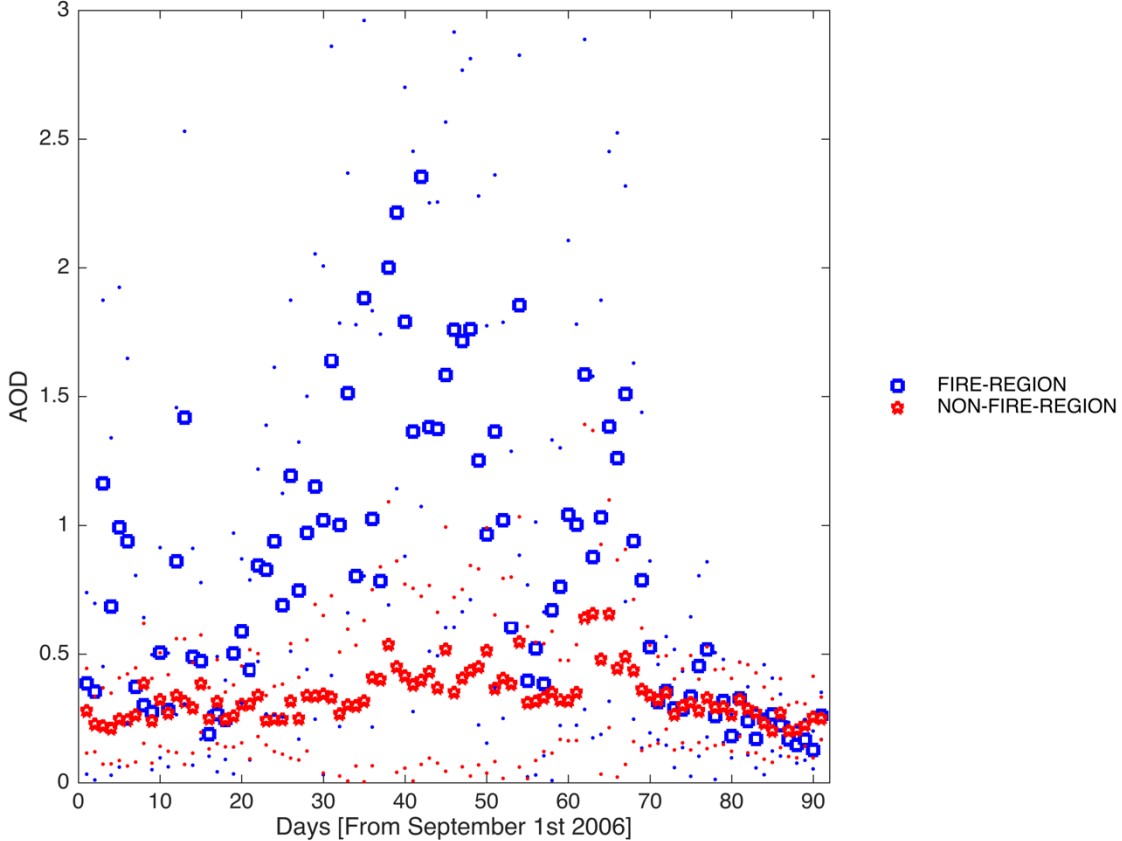


**Figure 3a,3b:** Time series of measured CALIPSO extinction heights over the fire constrained (A) and non
fire-constrained (B) regions as given **Figure 1**. Note that for the fire constrained region, the analysis (and
hence the data) has been extended for the period from September 3[rd] through November 9[th]. For both plots,
the dots correspond to the height of the column integrated backscatter at: 10% [red] (top), 30% [dark blue],
50% [yellow], 70% [black], and 90% [light blue] (bottom). The circles are computed daily means, while
dots are the computed daily standard deviation bands. There was no measurement over the region on
September 7[th], 8[th], 9[th], 11[th], 15[th], 16[th], 17[th], 18[th], 21[st], and October 10[th], 16[th], 20[th], 25[th], and 27[th].

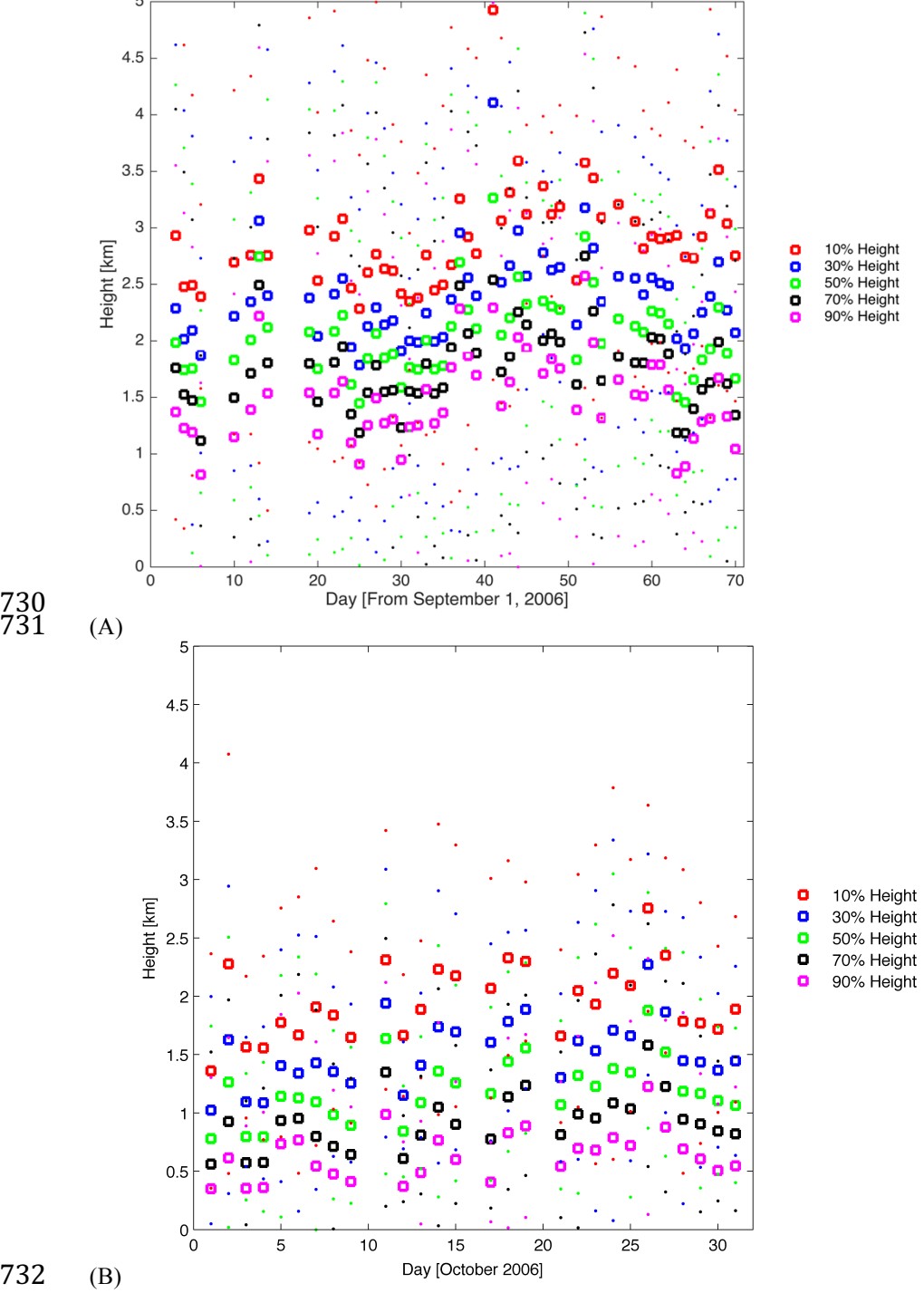

(A)
(B)
**Figure 4:** Time series of measured extinction height levels for the median heights (red circles and line) with
their corresponding +-1 standard deviation range (red dotted line), and respective middle-upper (blue), and
middle-lower (yellow), are given below. The best fitting modeled heights for the median daily boundary
layer height of 1000m are given as black x's, and are found to be respective FRP enhancements of 1.0, 1.2,
and 1.4. The best fitting modeled heights for the low daily boundary layer height of 700m are given as
black +'s, and are found to be respective FRP enhancements of 1.0, 1.1, and 1.2.  The best fitting modeled
heights for the high daily boundary layer height of 1300m are given as black o's, and are found to be
respective FRP enhancements of 1.0, 1.4, and 1.8.

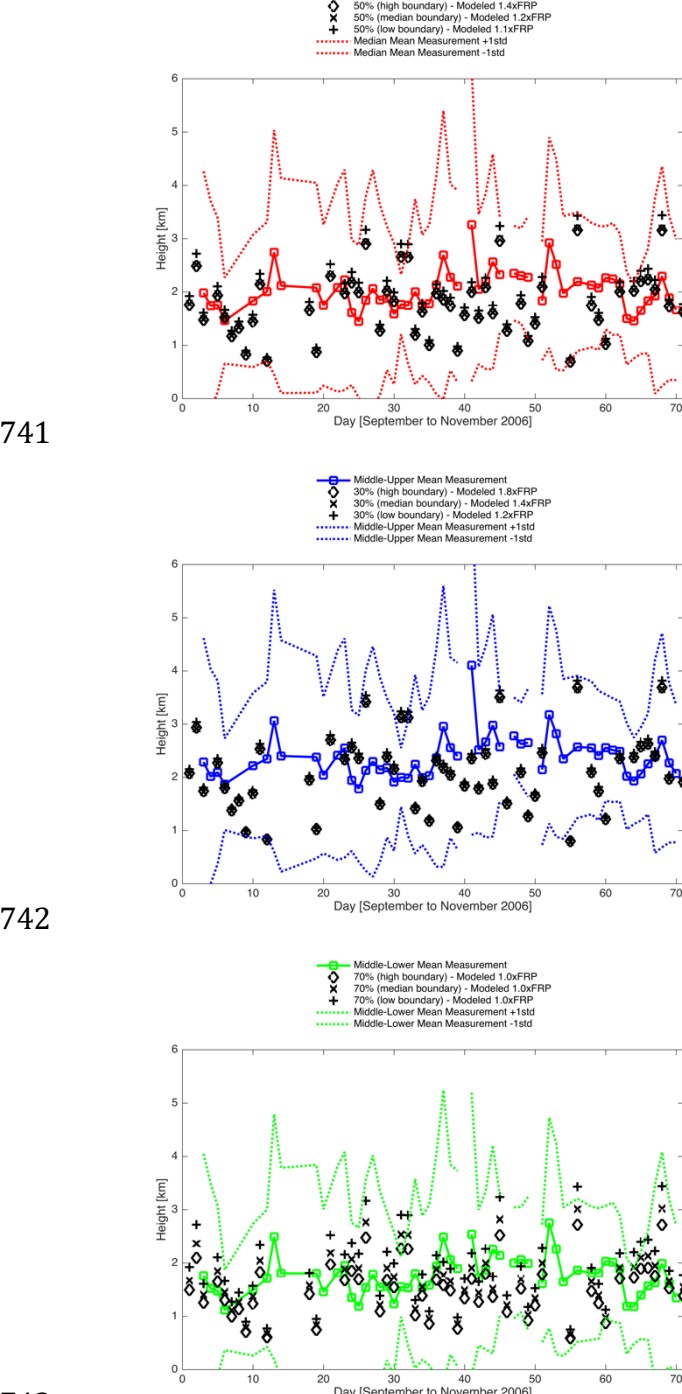


