# Peer review of "Vertical distribution of aerosols over the Maritime Continent during El Nino"

_Atmospheric Chemistry and Physics, 2016_

## Referee Comment (RC1) · Anonymous Referee #2 · 15 Mar 2017

The paper titled "Vertical distribution of aerosols over the Maritime continent during El Nino" authored by Jason Blake Cohen, Daniel Hui Loong Ng, Alan Wei Lun Lim, Xin Rong Chua presented the vertical structure of aerosols and fire radiative power during a large-scale fire events over the Maritime continent. They have observed that simple plume rise model underestimate the aerosol layer heights during the fire events and the fire radiative power is underestimated by $\sim$20%. As authors mentioned, generally satellites and models have significant bias in simulating aerosol properties, especially the vertical distribution. Authors didn't mention about the aerosol-retrieval uncertainties over the land, especially during large-scale fire events. How good are MODIS and MISR retrievals over Southeast Asia? What is the mean AOD observed during the fire events? Whether authors validated satellite and plume-rise model prior to this study? Why did authors use AOD and FRP from two different satellites instead a single

satellite? How does the atmospheric stabilization due to direct effect of aerosols affect the vertical transport of aerosols? By changing the fire radiative power, authors are able to reasonable reproduce the central height of the plume but failed to reproduce the other features. This may be attributed to several other reasons, as such this simple exercise doesn't warrant publication.

Minor comment The title of the manuscript is not apt (misleading also) for this study. This has nothing specific to about El Nino, general to all large-scale fire events. Line 16: Authors mentioned that "our results are significantly different from what others are using". However, it is hard to find any discussion on this topic in later sections. Provide references to justify the statement. Line 39-46: Poor clarity and readability Line 58: hydroscopicity or hygroscopicity? Line 101-107: Rewrite the sentence. Message is not clear. Line 110: What is the "reasonable approximation" mentioned here? Line 128: What do you mean by "best solution at 0.55 micron"?

Due to lack of clarity and inadequate data to support the scientific conclusions reported in this study, I would not recommend this manuscript for the publication in Atmospheric Chemistry and Physics

---

## Referee Comment (RC2) · Anonymous Referee #1 · 18 Mar 2017

This manuscript presents analysis of measurements and modeling of vertical distributions of aerosols during post El-Nino fire season. There is no doubt that vertical distributions of aerosols are of great importance and large uncertainties exist due to lack of measurements and thus poorly constrained model results. The authors claim that their results are significantly different from other previous studies. However, the analysis are mostly based on statistical analysis of vertical CALIPSO data and simple application of plume rise model. It is well known that large uncertainties remain in plume rise modeling, and many efforts have been made to improve plume rise modeling and improve fire emissions. The used CALIPSO data may also have large uncertainties during fire events and necessary verification are not provided. Besides, the authors overuse some words like "significant" (used more than 20 times) without showing how significant. The authors conclude that significant amount of aerosol mass exists in the

free troposphere using assumption that boundary layer height is 1000m in Singapore. However, this assumption might not be reasonable during severely polluted case like fire, since previous studies found that aerosols can stabilize boundary layer height. The authors should be very careful when use some words. I would not recommend publishing this work before the authors can include more comprehensive evaluation of the used methods and show more interesting results.

---

## Author Comment (AC2) · 1 Apr 2017

Response to Reviewer Number 1:

We thank reviewer number 1 for his/her comments. There are some constructive comments. Especially in terms of the uncertainties in the boundary layer heights being made more prominent and in terms of uncertainties in the lidar measurements being more carefully examined. Furthermore, there are multiple suggestions for improving wording, and for making clear when statistical significance is used, as compared to other comparisons. We appreciate these suggestions.

However, we want to emphasize that in fact, this work is a significant advance compared to other works in the literature. We hope to demonstrate that this work is up to date in terms of calibration, new in scope, and provides some major advances of

interest to the larger community:

One of the most comprehensive papers with respect to CALIOP over Southeast Asia, is given by Lee et al. [2016] (plus an entire team associated with NASA). They published a paper using CALIOP data over Southeast Asia, using slightly less error quantification for high ice clouds found in the tropics [although they may have done this from the way they configure figure 3, but it is not explicitly mentioned], which we have employed. Additionally, they used products that we knew are less reliable, such as SSA, and assumed that they were extendable functionally as the backscatter ratio goes (which is not necessarily a good assumption, but is certainly a much weaker assumption than we use in equating heights to backscatter). We made sure to stick to only the most reliable product, backscatter, as is also demonstrated in more depth by Rogers et al. [2009] and Hostetler [2008]. Furthermore, we performed the exact thing that these papers identified as the major weakness of CALIOP data, a methodology of how to combine the spatially-disparate paths, into a useful contiguous product.

As can be seen in the literature, many other papers use CALIOP without any validation at all [Sugimoto et al., 2015], or actually use it to validate models, which are known to themselves be highly inaccurate. This can be seen by the many papers put put by Jeff Reid and his team, whereby NAAPS (A modeling system) is used to validate CALIOP [Campbell et al., 2013]. A quick look at their most important validation for the NAAPS model (supplemental figure 6) shows that it performs less well than the modeling system (Cohen, 2014; Cohen and Wang, 2014; Cohen et al., 2017) over this region during the biomass burning season, since the annual average values still perform only as well, and due to the biases included within. Furthermore, as already demonstrated in the paper's references, the results are very comparable with findings from Lin Neng-Hui's group in Taiwan [Lin et al, 2014, 2013, etc.], and the AD-Net [Sugimoto et al, 2014] who are making observations with on-the-ground lidar at multiple places within the Northern portion of Southeast Asia and Greater East Asia. Hence, it should be completely reasonable to assume that the validation against the results

and findings from the underlying measurements, upon which those results are based, should be sufficient validation.

In an ideal world, there would be more validation on the ground in this region. However, this team, and many others are work to continue to improve this situation as best as possible. If there are any specific comparisons that you could recommend to continue to demonstrate the uniqueness of this result and its strength, compared to the other papers currently available, please let us know. From what we can see, this is the only paper that is comprehensive in its analysis on a day-by-day basis over a long-term such fire event, looking carefully over its total spatial extent over this region, not merely hunting for or tracing a few daily events, or randomly sampling over a very long spatial/temporal extent which is mixing both fire and non-fire type of atmospheric conditions.

Many of the other comments in terms of validating the errors of the models and measurements are mentioned in the other response and will not be repeated here. However, we take these suggestions very seriously and look forward to work with the reviewer to address their concern of the newness and significance of the results.

Finally, we agree that more information can be provided, and hope to do this by producing a probability distribution function of the results, on a day-by-day basis as well as over the entire month, for both the measurements and the models, and will include a summary as a new figure, with the remainder placed in appendices. We hope that this can address and demonstrate the significant importance, especially given the fact that the actual, stabilized boundary layer, may be lower than the 1000m as measured over Singapore.

References: Campbell, J.R., Reid, J.S., Westphal, D.L., Zhang, J.L., Tackett, J.L., Chew, B.N., Welton, E.J., Shimizu, A., Sugimoto, N., Aoki, K., Winker, D.M. (2013) Characterizing the vertical profile of aerosol particle extinction and linear depolarization over Southeast Asia and the Maritime Continent: The 2007–

2009 view from CALIOP, Atmospheric Research, 122, March 2013, 520–543, http://dx.doi.org/10.1016/j.atmosres.2012.05.007.

Hostetler, C., Hair, J., Liu, Z.Y., Ferrare, R., Harper, D., Cook, A., Vaughan, M., Trepte, C., Winker. D. (2008) Validation of CALIPSO Lidar Observations Using Data From the NASA Langley Airborne High Spectral Resolution Lidar (Retrieved from: https://ntrs.nasa.gov/archive/nasa/casi.ntrs.nasa.gov/20080014234.pdf)

Lee, J., Hsu, N.C., Bettenhausen, C., Sayer, A.M., Seftor, C.J., Jeong, M.J., Tsay, S.C., Welton, E.J., Wang, S.H., Chen, W.N. (2016) Evaluating the Height of Biomass Burning Smoke Aerosols Retrieved from Synergistic Use of Multiple Satellite Sensors over Southeast Asia, Aerosol and Air Quality Research, 16: 2831–2842 doi:10.4209/aaqr.2015.08.0506

Rogers, R.R., Hostetler, C.A., Ferrare, R.A., Hair, J.W., Obland, M.D., Cook, A.L., Harper, D.B., Swanson, A.J. (2009) Validation of CALIOP Aerosol Backscatter and Extinction Profile Products Using Airborne High Spectral Resolution Lidar Data (Retrieved from: http://cimss.ssec.wisc.edu/calipso/meetings/cloudsat_calipso_2009/Posters/Rogers.pdf)

Sugimoto, N., Nishizawa T., Shimizu A., Matsui I., Jin Y. (2014) Characterization of aerosols in East Asia with the Asian dust and aerosol lidar observation network (AD-Net) Proc. SPIE 9262 92620K

Sugimoto, N., Shimizu, A., Nishizawa, T., Matsui, I., Jin, Y., Khatri, P., Irie, H., Takamura, T., Aoki, K., Thana, B. (2014) Aerosol characteristics in Phimai, Thailand determined by continuous observation with a polarization sensitive Mie–Raman lidar and a sky radiometer, Environmental Research Letters, 10, 6.

---

## Author Response (AR1)

[revised manuscript text omitted]

We thank reviewer number 1 for his/her comments. There are some constructive
comments. Especially in terms of the uncertainties in the boundary layer heights being
made more prominent and in terms of uncertainties in the lidar measurements being more
carefully examined. Furthermore, there are multiple suggestions for improving wording,
and for making clear when statistical significance is used, as compared to other
comparisons. We appreciate these suggestions.

However, we want to emphasize that in fact, this work is a significant advance
compared to other works in the literature.  We hope to demonstrate that this work is up to
date in terms of calibration, new in scope, and provides some major advances of interest
to the larger community:

One of the most comprehensive papers with respect to CALIOP over Southeast
Asia, is given by Lee et al. [2016] (plus an entire team associated with NASA). They
published a paper using CALIOP data over Southeast Asia, using slightly less error
quantification for high ice clouds found in the tropics [although they may have done this
from the way they configure figure 3, but it is not explicitly mentioned], which we have
employed. Additionally, they used products that we knew are less reliable, such as SSA,
and assumed that they were extendable functionally as the backscatter ratio goes (which
is not necessarily a good assumption, but is certainly a much weaker assumption than we
use in equating heights to backscatter). We made sure to stick to only the most reliable
product, backscatter, as is also demonstrated in more depth by Rogers et al. [2009] and
Hostetler [2008]. Furthermore, we performed the exact thing that these papers identified
as the major weakness of CALIOP data, a methodology of how to combine the spatially-
disparate paths, into a useful contiguous product.
As can be seen in the literature, many other papers use CALIOP without any
validation at all [Sugimoto et al., 2015], or actually use it to validate models, which are
known to themselves be highly inaccurate. This can be seen by the many papers put put
by Jeff Reid and his team, whereby NAAPS (A modeling system) is used to validate
CALIOP [Campbell et al., 2013]. A quick look at their most important validation for the
NAAPS model (supplemental figure 6) shows that it performs less well than the modeling
system (Cohen, 2014; Cohen and Wang, 2014; Cohen et al., 2017) over this region during
the biomass burning season, since the annual average values still perform only as well,
and due to the biases included within. Furthermore, as already demonstrated in the
paper's references, the results are very comparable with findings from Lin Neng-Hui's
group in Taiwan [Lin et al, 2014, 2013, etc.], and the AD-Net [Sugimoto et al, 2014] who
are making observations with on-the-ground lidar at multiple places within the Northern
portion of Southeast Asia and Greater East Asia. Hence, it should be completely
reasonable to assume that the validation against the results and findings from the
underlying measurements, upon which those results are based, should be sufficient
validation.
In an ideal world, there would be more validation on the ground in this region.
However, this team, and many others are work to continue to improve this situation as
best as possible. If there are any specific comparisons that you could recommend to continue to demonstrate the uniqueness of this result and its strength, compared to the other papers currently available, please let us know. From what we can see, this is the only paper that is comprehensive in its analysis on a day-by-day basis over a long-term such fire event, looking carefully over its total spatial extent over this region, not merely hunting for or tracing a few daily events, or randomly sampling over a very long spatial/temporal extent which is mixing both fire and non-fire type of atmospheric conditions.

Many of the other comments in terms of validating the errors of the models and measurements are mentioned in the other response and will not be repeated here. However, we take these suggestions very seriously and look forward to work with the reviewer to address their concern of the newness and significance of the results.

Finally, we agree that more information can be provided, and hope to do this by producing a probability distribution function of the results, on a day-by-day basis as well as over the entire month, for both the measurements and the models, and will include a summary as a new figure, with the remainder placed in appendices. We hope that this can address and demonstrate the significant importance, especially given the fact that the actual, stabilized boundary layer, may be lower than the 1000m as measured over Singapore.

determined by continuous observation with a polarization sensitive Mie–Raman lidar and
a sky radiometer, Environmental Research Letters, 10, 6.

Response to Reviewer Number 2:

We thank reviewer number 2 for his/her comments. There are some constructive comments. Additionally, some of the comments show indirectly that the wording should be improved for clarity, as part of the structure of the way the research was organized was possibly misunderstood.

However, we strongly urge the reviewer to be more careful, and to actually read the paper to the end. The reason being that a few of the points made, including stating that the data was "inadequate" to "support the scientific conclusions", were actually already addressed in the paper, through figures and tables.

These details will be given below in the point-by-point response. Reviewer comments are preceded by RC: and are *italicized*, while the author responses are preceded by AC:

RC: *"Authors didn't mention about the aerosol-retrieval uncertainties over the land, especially during large-scale fire events"*
AC: This has been mentioned in other papers we have cited and already performed over this region (Cohen, 2014; Cohen et al., 2017). However, for clarity the values will be added into this paper. There are two issues, the first with cloud cover, in which a bias may exist because extremely high AOD conditions (AOD>2) are frequently flagged as clouds. The second is the error itself over land, which can go as high as 15%. However, as you can see from the figures and the tables, the difference between the "fire region" and the "non-fire region" is significant at values where the errors are much larger than 15%. This has already been factored in, but will be made clear. The terms of the issues of the bias by cloud screening reducing the measured AOD is if anything a stronger supporting issue for the results given here, since it would make the difference between the "fire region" and the "non-fire region" even larger. It is also the reason why MISR was used for the initial definition of the two regions, since its ability to cloud clear is better than MODIS over this region. It is furthermore a point of interest when analyzing the CALIOP data, which has some ability to distinguish between aerosol and cloud. There is discussion of this in Section 2.2 and it can be expanded.

RC: *"How good are MODIS and MISR retrievals over Southeast Asia?"*
AC: This has been mentioned in other papers we have cited and already preformed over this region (Cohen, 2014; Cohen et al., 2017). In general, these products need to be used carefully, but the relationships found, when applying the techniques outlined here and in previous work, match extremely well with ground measurements from AERONET, within the errors of the 15% stated above. Additionally, model results forced with these measurements, have been shown to match very well with NOAA, Chinese, and other networks downwind throughout Southeast and East Asia. Additional comparisons about this can be included. And these matches have been demonstrated over not only the time period of this paper, but for the past decade. There are many errors using the satellite data, but the errors are sufficiently small as to not impact the analysis when looking at the events outlined here, due to their intense magnitude and spatial extent. There is discussion of this in Section 2.2 and it can be expanded.

AC: This was clearly something that the reviewer missed. I urge the reviewer to carefully
look at **Figure 3** and the text on lines 132-138. I am attaching them for reference:

**Figure 3:** Time series of daily averaged measured AOD over the fire-constrained regions of the Mariti
Continent [blue], and the non fire-constrained regions of the Maritime Continent [red], as given in **Figu**
**1**. Circles are computed daily mean values, while dots are computed daily standard deviation bands.

[Figure]

*et al.*, 2007]. On the other hand, while some grids are contaminated, the sheer spatial distance of the plume
and the fact that the overwhelming majority of atmospheric aerosols during this time of the year are due to
fires, means that there is no significant bias in the overall statistics of the measured AOD [*Cohen*, 2014], as
observed by looking at the spatially averaged MODIS AOD and statistics over the fire-constrained and non
fire-constrained regions (**Figure 3**). The AOD is significantly higher (p<0.01) over the fire-constrained
region, making the findings consistent with the approach employing the 12 years worth of MISR
measurements.

*RC: "Whether authors validated satellite and plume-rise model prior to this study?"*

AC: Again, it seems that the reviewer did not carefully read the previous works of the
author, in which careful validation of the measurements and models have already been
performed. Perhaps, for clarity, additional paragraphs can be added to this work to further
enhance this, without repeating verbatim. Specifically, a quick summary of the
performance under the local conditions in Southeast Asia, which seems to be the
reviewer's primary concern.
*RC: "Why did authors use AOD and FRP from two different satellites instead a single*
*satellite?"*
AC: We did not. All of the AOD and FRP measurements used in the statistics and figures
(except for figure 1) are from MODIS. We used MISR only to constrain in space and time
the domain that was influenced by the fires "fire region" and no influenced by the fires
"non-fire region". This is explained in detail in **sections 2.1 and 2.2**. The explanation
includes the major rationale, since MISR cloud clears better over this region, but has a
lower frequency of measurement.
*RC: "How does the atmospheric stabilization due to direct effect of aerosols affect the*
*vertical transport of aerosols?" and "This may be attributed to several other reasons"*
AC: As mentioned in the article, there is an atmospheric stabilization due to both the
DIRECT EFFECT and the SEMI-DIRECT EFFECT. There is literature to support this,
some of which has been cited in the draft. It is an excellent question, and a question we
are currently tackling. Additionally, we talk about the issue of localized convection also
not being properly resolved. The fact of the matter is that, at the present time, there are
hundreds of papers being published with regional models (i.e.: WRF-CHEM) and global
models (i.e. GEOS-CHEM), which do not also address these issues at all. In fact, they are
not even designed to allow for communication between the chemistry and the
meteorology.
The rationale of this work was to follow their approach, which is a currently
accepted approach in the community, and to do an extremely comprehensive study. There
is currently no other paper that has analyzed more than 10,000 daily data points of
CALIOP measurements, and run a model jointly with more than 3000 MODIS daily fire
hotspots and meteorological measurements, over this region of the world, that we can
find. In fact, the papers that are generally cited over this region of the world do not even
mention what methodology they use to analyze the measured data, nor details of which
versions of the data they are using. Frequently, they make mere comparisons with
CALIOP or AOD image files, without carefully looking at the data and making sure it is
of high quality. I have even seen such articles published in this prestigious journal.
*RC: "This has nothing specific to about El Nino, general to all large-scale fire events."*
AC: This is not true, and the reviewer knows this. While this was a large-scale fire event,
it was very special. The Monsoon has generally arrived in October, and hence October is
usually the transition from the burning season to the rain season. As such, the
meteorology was completely different during this period of time. It was heavily
influenced by El Nino. The length of the burning season, the wetness of the soil, the large-scale meteorological flow, and such, were all not typical in this year. It is an interesting idea, however, to expand upon such and to compare the differences between El Nino and non El Nino years, however, the amount of data analysis to be required would be huge. In fact, this is a common mistake made by other authors in the past in this region of the world, and one of the major reasons why their results do not necessarily compare as well against measurements from AERONET, NOAA, Chinese measurement networks, etc.

RC: *"Line 16: Authors mentioned that "our results are significantly different from what others are using". However, it is hard to find any discussion on this topic in later sections. Provide references to justify the statement."*
AC: Thank you. While there are already some references, more will be added, and the discussion will be extended as well.

RC: *"Line 39-46: Poor clarity and readability"*
AC: This will be addressed. Thank you.

RC: *"Line 58: hydroscopicity or hygroscopicity?"*
AC: Hygroscopicity, thank you.

RC: *"Line 101-107: Rewrite the sentence. Message is not clear."*
AC: Thank you. This will be addressed.

RC: *"Line 110: What is the "reasonable approximation" mentioned here?"*
AC: References and some more detailed explanations of the physics will be added here. It is based on the physical relationship between backscatter and the aerosol optical properties, which are reasonably uniform when considered over the thousands of kilometers extent of the plume as a whole.

Changes to the Supplement

**Supplement:**

**Detailed Methodology**

The buoyancy flux parameter ($F_B$) **Equation A1** is a function of the temperature difference between the air ($T_A$) and the fire ($T_F$), the vertical motion of air (v) and the size of the fire, d (here always measured at 1km$^2$ in this work).

$$F_B = gv\frac{d^2}{4}\left(\frac{T_F - T_A}{T_A}\right)$$

(A1)

The buoyancy flux parameter has been found empirically to demonstrate whether the plume rise is buoyancy or momentum dominated. Under stable atmospheric conditions [Stone and Carlson, 1979], where the atmospheric lapse rate is ($L_A = \frac{\Delta T}{\Delta Z} < -5$), for a buoyancy dominated plume, (defined as where the difference between $T_A$ and $T_F$ is given in **Equation A2b1**), the plume rise height ($\Delta h$) is given by **Equation A2b2**, where (U) is the horizontal wind magnitude.

$$(T_F - T_A) > 0.01958 T_F \sqrt{v}$$

(A2b1)

$$\Delta h = 2.4\left(\frac{F_B}{.02U}\right)^{1/3}$$

(A2b2)

Whereas, for a momentum dominated plume (where the difference between $T_A$ and $T_F$ is less than the right hand side of **Equation A2b1**), the height rise is given by **Equation A2b3**.

$$\Delta h = 1.5\left(\frac{\frac{v^2 d^2 T_A}{4} \frac{T_A}{T_F}}{\sqrt{.02U}}\right)^{1/3}$$

(A2b3)

On the other hand, under unstable atmospheric conditions (where $L_A > -5$), and where the plume rise is buoyancy dominated, the plume rise height is given by either **Equation A2b4** when $F_B > 55$ or **Equations A2b5, A2b6** when $F_B < 55$ [Woodward, 2010].

$$X^* = 14F_B^{\frac{5}{8}}$$

(A2b4)

$$X^* = 34F_B^{\frac{2}{5}}$$

(A2b5)

$$\Delta h = 1.6\frac{F_B^{\frac{1}{3}}(3.5X^*)^{\frac{2}{3}}}{U}$$

(A2b6)

**Supplemental Figure 1:** PDFs (x-axis is the height in km, and the y-axis is the probability distribution) of the monthly aggregated backscatter heights of the 10% [red] (top), 30% [dark blue], 50% [yellow], 70%

[light blue], and 90% [black] levels. Note that there were no measurements on the 10th, 16th, and 20th.

[Figure]

**Supplemental Figure 2:** Map of the monthly averaged MODIS AOD over the Maritime Continent. The
day-to-day statistics are given in **Figure 3**. Regions in white have 0 valid AOD measurements throughout
the entire time period, due to cloud cover.

---

## Referee Report (RR1)

**Review of "Vertical distribution of aerosols over the Maritime Continent during the El Nino" by Jason Blake Cohen et al.**

The manuscript uses satellite-derived AOD to spatially and temporally constrain the sampling of smoke aerosols with an aim to examine the aerosol vertical distributions, measured from CALIOP, over the maritime continent during the 2006 El Nino. The observed aerosol vertical distributions were then used to compare with the results of a simple plume rise model. The study provides some insights into the aerosol signature in terms of vertical distribution during El Nino conditions and the limitations of plume rise models. But additional evidences and analysis are required to support the conclusions. Several major comments have to be addressed. Furthermore, the manuscript readability and clarity has to be improved before the publication in ACP.

**Major comments:**

1. More elaborations and descriptions are required for CALIPSO data processing. Which version and level of CALIOP product? Is each individual measurement under cloud-free conditions? If yes, which cloud mask data was used? How many samples in total? What is the threshold value of extinction form CALIOP data (please consider the daytime background solar illumination by Winker et al., 2013)? In addition, an analysis of the uncertainties of the CALIOP-derived vertical aerosol extinction, in particular over this region, is needed.
2. The effects of the uncertainty in boundary layer depth need to be considered. The authors simply use the 1000 m to approximate the boundary layer height. Assuming the boundary layer has +/- 300 m uncertainties during the CALIPSO overpass, which is totally possible, what are the uncertainties of the percentage of free atmosphere aerosol estimated by your method? When taking this into account, how does your result compare with previous studies?
3. The author uses aerosol-induced in-situ stabilization as a possible explanation to the underestimation of plume height by model. But the rationale seems problematic. The model does not account for the effect of aerosol-induced stabilization which actually happens in the real atmosphere. The stabilization causes weaker buoyancy, thus lower plume height. Therefore the model that misses such stabilization should overestimate, not underestimate, the plume height.
4. The comparison between model and observation is insufficient. The observation misses 3 days and the model misses 9 days with only 18 days left. This is rather a small sample. Since the fires lasts from September to November, it is worthwhile to expand the analysis to September and November. In addition, the analysis is primarily limited to the monthly averages. The authors do show the comparisons in daily basis in Figure 4, but do not analyze them. In particular, the three special days mentioned in section 3.2 are good example cases to analyze in order to shed more light on the observation-model comparisons. Such more comprehensive analysis is very worthwhile in order to support the conclusions of how to reduce model bias which is actually not well examined or indicated in the manuscript.

**Specific comments:**

Line 15: "measurements and modeling". Please specify which measurement and which model.

Line 16: "underestimated" by what?

Line 51: Sentence not readable

Line 53 ~ 54: "underestimation" in "spatial, and temporal distribution"?

Line 60: Change "show" to "shown"

Line 75: Show full name of "CALIOP"

Line 77: Show full name of "SSA"

Line 77: "go with each pass". Is it scientific language?

Line 82: grammar error

Line 85: Show full name of "MISR"

Line 157: Please provide the reference.

Line 158: delete one "are"

Line 162~167: Show full name of "AERONET", "NOAA","RMS", "RCP","GDED"

Line 167: what is R2 statistic?

Line 204~206: Please provide new plot to show them more directly.

Line 218: Show full name of "BC"

Line 272: add "of" after "and"

Line 285~286: Why not examining the hypothesis by looking at precipitation data from A-Train?

Line 290: Show full name of "MERRA"

Line 360-361: "vertical distribution" is not a parameter to be "estimated".

Figure 1: What does the color stand for? Please add a title to the colorbar.

Figure 4: The comparison is really not readable. Please make it more clear.

---

## Author Response (AR2)

**Response to Reviewer Number 3:**

We thank reviewer number 3 for his/her comments. In particular, we will focus on the data quality issues here. We also will focus on the expansion of the overall amount of data used, so as to reinforce the validity and uniqueness of this approach. We greatly appreciate all of the depth and complexity, and hope that our hard work has addressed the various suggestions improvements recommended.

The Reviewer's comments in italics, while our specific responses are in plain text.

*"1. The plume height analysis entirely depends on CALIPSO extinction profiles and vertical feature mask product. However, both these two products bare uncertainties that could contaminate the results. Especially that the feature classification is based on lidar ratio which is highly uncertain. The author should provide evidence that these data are validated against ground based lidar for the region and period of interest."*

We have carefully analyzed the MPL data from the Singapore station, which exists within the "Fire-Region" in this study. We find that throughout the period, the general average mean extinction height varies from 1.8km through 2.1km, which is consistent with the statistical results found using CALIPSO. Hence, we believe we have done at least as well as others, with respect to validating the variables used, over this region. We have added in Supplemental Figure 3 and the following text:
"These results are supported by the statistical values of aerosol heights measured by the MPL station in Singapore throughout the period from September 1 to November 30 (**Supplemental Figure 3**), which are found to range from 1.6km to 2.4km. While there were no ground-based lidar measurements available in 2006, the year 2015 was another very strong El-Nino year which impacted Singapore with severe downwind aerosols from burning sources, and closely resembled 2006."

*"2. The fire region is selected according to MISR AOD. While I agree that MISR cloud screen might be better than MODIS, its sampling is rather poor over low latitudes. This means that many small scale plumes may not be captured by MISR at all. I suggest the authors compare both MODIS and MISR and maybe also OMI which is good in measuring absorbing aerosols to better determine the fire region or to confirm their defined region."*

Due to the extensive spatial distribution of the fires, there are almost no visible plumes over this region. They all intermix, and effectively act as a single plume. This is clearly mentioned in the text. In previous work already cited here, we have found that MISR performs better than MODIS in this region of the world, when compared against AERONET. However, we have also used MODIS in this study, as demonstrated in **Figure 2** and **Supplemental Figure 2**. To validate our assertion, we have found that the actual spatial area of cloud-free measurements provided by MODIS is considerably less than MISR. However, the fact that a similar conclusion is made, that the "Fire-Region" has a much higher AOD than the "Non-Fire-Region", means that these different platforms are at least achieving a similar large-scale result.

*"3. The authors state that the CALIPSO statistics is based on more than 10,000 profiles.*
*In my opinion, it is not the number of profiles that matters but the number of plumes or*
*fire events that the sampling represents. Usually one plume or event may contain tens or*
*even hundreds of profiles. So are these 10,000 profiles complete or representative or the*
*entire biomass burning season?"*

The CALOIP passes are as representative of the total fires as is possible. Every single
measurement that intersects the geographical region of interest, is used. Additionally, due
to the frequency of the passes, there is enough sampling done to account for any localized
events in space or time that may not be representative of the overall fire statistics. To
address the issue of representativeness in time, we have expanded the paper's analysis to
encompass the entire fire season the paper, now using three months of measurements,
from September $1^{st}$ to November $30^{th}$. The fire season was found to extend from
September $3^{rd}$ through November $9^{th}$. And within this period as a whole, the findings are
similar, with the results not significantly changing. Please see the updated results in
**Figure 2**, **Figure 3a**, and **Table 1**.
In fact, the only observed change is that the measured heights are slightly lower when
also incorporating in the data from September $3^{rd}$ to September $30^{th}$ and from November
$1^{st}$ to November $9^{th}$. This is consistent with the fact that while both September and
November recorded as significant in the past work, that the difference in the measured
AOD between the fire-region and non-fire region during this time was smaller than
during the October peak. Again, this reinforces the robustness and inclusivity of this
approach.
The first conclusion, as already mentioned in this work and previous work, it is
impossible to talk about "number of plumes" or "number of fire events". The entire
region is burning, or such a significant amount of the region, that in effect, there is just a
single giant plume as far as the atmosphere is concerned. The entire idea of counting
individual plumes is outdated, and not really possible. In the atmosphere, due to the
amount of fires, and their density, the overall distribution behaves as a "single massive
plume".

*"4. The conclusion that fire power is underestimated is solely based on matching the*
*plume rise model with CALIPSO observed mid plume height. However, the model still*
*shows quite different low and high plume heights. Given that the latter two is also related*
*to emission power, the conclusion seems unsound."*

We believe that the best fit matches with the data for other plume heights are also
mentioned in the text. In fact, we specifically stated that a nearly doubling of the radiative
power is required to match with the middle-upper height. We additionally talk about how
there are other, higher-order and non-linear feedbacks, that impact the system, that are
beyond the scope of this analysis, but are consistent with the findings here, especially for
the top and bottom values. The basic point is that even a scaled plume-rise approach will
not be able to accurately reproduce the top or bottom of the plume, under the conditions
observed in this environment.

This is especially so in the cloudy tropics. Under such intense radiation, strongly
absorbing smoke aerosol has significant impacts on the direct effect (near the bottom of
the column) and the semi-direct effect (near the top of the column). Furthermore, local
convection may also impact the plume rise height. This is already mentioned in the paper.
*"5. I still have some problem with the claim that the result of the current paper is very*
*different from previous studies. This is also the main issue raised by the first round of*
*reviewers. The authors cited Tosca et al. (2011) who stated that the plumes are mostly*
*confined within the boundary layer. The Tosca et al. results are based on the entire 2001-*
*2009 period rather than just October 2006, therefore not directly comparable with the*
*current study. The authors need to be more careful about their statement and conclusion*
*and provide direct comparisons with previous research."*
First of all, the majority of fires occurring during the period from September through
November, occur in this region of the world. Secondly, 2006 was specifically chosen, as
it was one of the years with the most available amounts of smoke events, due to the
dryness associated with El-Nino. This was also clearly mentioned in Tosca et al. (2011).
In fact, their paper also specifically pointed to October 2006 as a case study, and hence a
direct comparison is indeed able to be made. This is why we have extended our analysis
from September $1^{st}$ to November $30^{th}$, so that we can capture the entire breadth of the fire
season. In addition, other such papers have also now been cited in the latest update,
including: Campbell, et al. (2013); Lee, et al. (2016); Sugimoto, et al. (2014a); and
Sugimoto, et al. (2014b). It is true that there are not many papers with respect to this
region, and furthermore it is true that none of the available other works have gone into the
depth and clarity with how they have analyzed the data, and hence, it may not be possible
to find any more closely-related work.
*"6. Lines 162-163: is the resolution 10km or 1km? Also I suggest the authors change the*
*"x" in "10kmx10km" to the real multiplication signs by inserting symbols."*
The symbol issue has been taken care of. Thank you for pointing this out. The resolution
is 10km for the AOD product and 1km for the FRP and fire temperature products.
*"7. Lines 183-185: please provide references that AOD and other products are*
*validated."*
Additional references indicating AOD validation have been included.
*"8. Lines 258-260: I don't understand this sentence. Please rephrase."*
This has been re-written. Thank you.
*"9. Line 289: "this work's underlying Kalman Filter plus variance maximization*
*inversely modeled fields", is this used in the plume rise model? If so, please describe*
*more specifically in the model description section."*

This is the spatial region that is used for defining the "Fire-Region" and the "non-Fire-
Region". This is now made more clear in the text.
*"10. Line 325: lower temperature should correspond to "lower emission factor" rather*
*than "higher aerosol emission factor"."*
We think it is much more complex than this, and hence disagree to making this change.
Lower temperature should correspond to a lesser amount of burnt material, and hence a
lower absolute emissions of total carbon to the atmosphere. However, sometimes at lower
temperatures, especially in the wet tropics, the actual mass of aerosol being produced is
higher. This has to do with the water content leading to less oxygen being available for
combustion, and the combustion occurring at a lower temperature. In the end, this leads
to more incomplete combustion, and hence more volatile species being produced, as well
as BC and OC emissions. Frequently at very high temperature, these species are more
highly oxidized and hence have less mass remaining in the aerosol phase. Thank you for
this interesting point, as it provides a future basis upon which to continue looking more
deeply into the topics raised here.
*"11. Line 400-401: This statement is too strong. Given the data quality problem and*
*model mismatch problem, I don't think the work "comprehensively quantifies …"."*
The suggestion has been addressed and the sentence re-worded.
*"12. The paper still has many grammar mistakes, such as verbs associated with plural or*
*single forms. Please double check"*
Thank you for pointing this out. A careful review has found some additional errors and
fixed them.

**Response to Reviewer Number 4:**

*"The manuscript uses satellite-derived AOD to spatially and temporally constrain the sampling of smoke aerosols with an aim to examine the aerosol vertical distributions, measured from CALIOP, over the maritime continent during the 2006 El Nino. The observed aerosol vertical distributions were then used to compare with the results of a simple plume rise model. The study provides some insights into the aerosol signature in terms of vertical distribution during El Nino conditions and the limitations of plume rise models. But additional evidences and analysis are required to support the conclusions. Several major comments have to be addressed. Furthermore, the manuscript readability and clarity has to be improved before the publication in ACP."*

Thank you very much for your deeply reflective and insightful comments. We have carefully parsed through them and worked our hardest to address them. We have done significantly more work and analytics, as well as extending the period of the data analyzed. Overall our same conclusions are found, but they are now strengthened. By extending the analysis to the entire fire season, not just the peak of the fire season, we have more conclusively exhibited our findings, and determined that the underlying points are still the same. We still find that the vast majority of the measured aerosol is in the free troposphere. Second, that the existing plume models are biased in terms of reproducing the lower portions of the plume, and are not capable of reproducing the extremes in the plume height. Third, that the underestimation of measured FRP leads to an improvement in being able to model the central characteristics of the plume, even given the uncertainty in average boundary layer characteristics, if FRP enhancements are applied piecemeal. However, fourth, that to model some essential statistics, such as the plume distribution, or the extreme plume height values, that fundamental changes to the models themselves will need to be made. We have also spent extensive time and care to re-write the paper and make it more readable.

*"1. More elaborations and descriptions are required for CALIPSO data processing. Which version and level of CALIOP product? Is each individual measurement under cloud-free conditions? If yes, which cloud mask data was used? How many samples in total? What is the threshold value of extinction form CALIOP data (please consider the daytime background solar illumination by Winker et al., 2013)? In addition, an analysis of the uncertainties of the CALIOP-derived vertical aerosol extinction, in particular over this region, is needed."*

Thank you very much for your detailed suggestions. A couple of paragraphs and sentences have been added addressing the data processing, cloud-conditions, masking data, and number of samples in total. In addition, we have thoroughly read your paper cited it, and included some specific points from it, since we believe that it strengthens the overall conclusion.
We do agree with you that a deeper study of aerosol extinction over this region is warranted, and possibly can follow-up in a future, more detailed analysis. However, at the present time, we do not consider or use the extinction data from CALIOP in this work, only from MODIS and MISR.

*"2. The effects of the uncertainty in boundary layer depth need to be considered. The*
*authors simply use the 1000 m to approximate the boundary layer height. Assuming the*
*boundary layer has +/- 300 m uncertainties during the CALIPSO overpass, which is*
*totally possible, what are the uncertainties of the percentage of free atmosphere aerosol*
*estimated by your method? When taking this into account, how does your result compare*
*with previous studies?"*
This is a very fair comment, especially given the uncertainty in boundary layer height in
the tropics. The analysis has been expanded to include this uncertainty band, and the
results have been correspondingly updated throughout the manuscript. The findings show
that the elevated levels during the October maximum are more significant than the entire
fire season, but that this difference is considerably smaller than between the fire-region
and the non fire-region. Furthermore, the difference between the boundary layer
uncertainty is also considerably smaller than between the fire-region and the non fire-
region. Hence, the results have been statistically strengthened by this analysis.
**Therefore, many thanks again for this suggestion**, even though it took a considerable
amount of time to properly implement.
*"3. The author uses aerosol-induced in-situ stabilization as a possible explanation to the*
*underestimation of plume height by model. But the rationale seems problematic. The*
*model does not account for the effect of aerosol-induced stabilization which actually*
*happens in the real atmosphere. The stabilization causes weaker buoyancy, thus lower*
*plume height. Therefore the model that misses such stabilization should overestimate, not*
*underestimate, the plume height."*
Actually, this is a very important point and it has been re-explained further in the text. We
agree with you that, at the surface, the aerosol effect reduces the buoyancy, by reducing
the incoming solar radiation. However, due to the large amount of highly absorbing
aerosols, it actually increases buoyancy near the top of the plume. And this increase is
further enhanced by the fact that once the aerosols are over the cloud top (as observed),
that this absorption is doubled. Hence, it serves the effect of reducing the heights near the
bottom, while simultaneously increasing the heights near the top. On the other hand, if
the surface fire radiative power were higher, say to the extent that most of the plume were
lofted to or above the cloud deck, which is what is observed, then this would not be the
case. The reduced buoyancy due to the aerosol direct effect is overcome near the bottom
by the additional heating. While on the other hand, the spread at the top would be
increased, due to the additional heating. Hence, a bias would occur, where the top of the
plume would be found to be biased slightly higher, which is what the measurements
seemingly demonstrate.
*"4. The comparison between model and observation is insufficient. The observation*
*misses 3 days and the model misses 9 days with only 18 days left. This is rather a small*
*sample. Since the fires lasts from September to November, it is worthwhile to expand the*
*analysis to September and November. In addition, the analysis is primarily limited to the*
*monthly averages. The authors do show the comparisons in daily basis in Figure 4, but*

*do not analyze them. In particular, the three special days mentioned in section 3.2 are*
*good example cases to analyze in order to shed more light on the observation-model*
*comparisons. Such more comprehensive analysis is very worthwhile in order to support*
*the conclusions of how to reduce model bias which is actually not well examined or*
*indicated in the manuscript."*
The analysis has been increased to the entire time period corresponding with the increase
in measurements and still constrained by the MODIS observations of being within the fire
season. There are now a total of 47 days in common to be analyzed. As is expected,
including the additional days has led to the mismatch between the model and the
measurements to be less large, but it has not changed the statistical significance, the sign,
or the overall value significantly. Details have been addressed in an updated **Figure 4** and
in the text. This includes some details of special days as well. As expected, the maximum
and most intense part of the fire season, October, has the largest mis-match. However, the
bias in the model mismatch, in particular for the median and lower plume heights, and the
large majority of the plume still being measured in the lower free troposphere are still
consistent across the entire fire season. Just less so over the entire fire season. This
additional work has strongly enhanced the overall results of the work.
*"Line 15: "measurements and modeling". Please specify which measurement and which*
*model."*
This has been modified and explained in more detail.
*"Line 16: "underestimated" by what?"*
This has been addressed.
*"Line 51: Sentence not readable"*
This has been addressed.
*"Line 53 ~ 54: "underestimation" in "spatial, and temporal distribution"? "*
This has been made clear.
*"Line 60: Change "show" to "shown" "*
Done.
*"Line 75: Show full name of "CALIOP" "*
Inserted at the first point CALIOP is mentioned.
*"Line 77: Show full name of "SSA" "*

Inserted at the first point SSA is mentioned.
*"Line 77: "go with each pass". Is it scientific language? "*
This has been updated to be more technical and precise.
*"Line 82: grammar error"*
This has been rewritten.
*"Line 85: Show full name of "MISR" "*
Done.
*"Line 157: Please provide the reference. "*
Done.
*"Line 158: delete one "are" "*
Done
*"Line 162~167: Show full name of "AERONET", "NOAA","RMS", "RCP","GDED" "*
Done
*"Line 167: what is R2 statistic? "*
Coefficient of determination. It relates the amount of variance observed in the response
variable by the test variable. This commonly used statistic has been defined more clearly
and in more depth.
*"Line 204~206: Please provide new plot to show them more directly. "*
**Figure 4** has been both updated to demonstrate the additional days of model results, as
well as being completely reformatted to make it clearer and easier to understand and
interpret. Thank you for the suggestion.
*"Line 218: Show full name of "BC" "*
Done.
*"Line 272: add "of" after "and" "*
Done.

*"Line 285~286: Why not examining the hypothesis by looking at precipitation data from*
*ATrain? "*
This has now been done in depth. One and a half paragraphs have been added to the
manuscript, as well as a Supplemental figure. The results support the previous hypothesis.
"One consistent rationale is that there was large-scale precipitation event at that time,
which in turn both increased aerosol removal and wetting of the surface. This in turn led
to lower temperature and FRP and correspondingly higher aerosol emissions factor on
these days. Overall, there is no apparent impact of day-to-day variability of measured
FRP driving observed variation in measured aerosol heights, and hence only high
confidence fire data is subsequently used.
To examine this hypothesis, the GPCP [Global Precipitation Climatology Project]
One-Degree Daily Precipitation Data Set of global precipitation has been employed to
study the amount and duration of rainfall over the fire-burning and non fire-burning
regions [Huffman et al., 2012]. A spatial/temporal analysis of this dataset, over both the
Fire Region and the No-Fire region confirms this hypothesis (**Supplemental Figure 4**)
Supp. Overall, there was considerably lower rainfall over the Fire Region than the No-
Fire Region, however, on all days that there was a decrease in AOD and FRP over the
Fire Region, there was a heavy Rainfall at the same time, or one or two days before. The
measurements have a correlation coefficient of -0.39 with a corresponding $p<0.01$. There
is no other statistically significant correlation found over any other combination of the
regions with any other combination of rainfall."
*"Line 290: Show full name of "MERRA" "*
Done. The sentence has also been re-written.
*"Line 360-361: "vertical distribution" is not a parameter to be "estimated". "*
This has been addressed.
*"Figure 1: What does the color stand for? Please add a title to the colorbar. "*
This has been added to the figure caption.
*"Figure 4: The comparison is really not readable. Please make it more clear. "*
This has been addressed in the new **Figure 4**.

[revised manuscript text omitted]

$$\Delta h = 1.5\left(\frac{\frac{v^2 d^2}{4}\frac{T_A}{T_F}}{\sqrt{.02U}}\right)^{1/3}$$

(A2b3)

On the other hand, under unstable atmospheric conditions (where $L_A > -5$), and where the plume rise is buoyancy dominated, the plume rise height is given by either **Equation A2b4** when $F_B > 55$ or **Equations A2b5, A2b6** when $F_B < 55$ [Woodward, 2010].

$$X^* = 14 F_B^{\frac{5}{8}}$$

(A2b4)

$$X^* = 34 F_B^{\frac{2}{5}}$$

(A2b5)

$$\Delta h = 1.6\frac{F_B^{\frac{1}{3}}(3.5X^*)^{\frac{2}{3}}}{U}$$

(A2b6)

**Supplemental Figure 1:** PDFs (x-axis is the height in km, and the y-axis is the probability distribution) of the monthly aggregated backscatter heights of the 10% [red] (top), 30% [dark blue], 50% [yellow], 70%

[light blue], and 90% [black] levels. Note that there were no measurements on the 10th, 16th, and 20th.

[Figure]

**Supplemental Figure 2:** Map of the monthly averaged MODIS AOD over the Maritime Continent. The day-to-day statistics are given in **Figure 2.** Regions in white have 0 valid AOD measurements throughout the entire time period, due to cloud cover.

[Figure]

**Supplemental Figure 3:** Statistical average of the aerosol heights measured by the Singapore MPL station from September 1 to November 30, 2015. This year was chosen since it is another El-Nino influenced high fire year, and has a somewhat similar physical, meteorological, and geographic aerosol extent as 2006.

**Supplemental Figure 4:** Time Series of Precipitation data from GPCP (dotted line) and AOD (dashed line)

from MODIS, averaged on a daily-basis over both the Fire Region (Red) and the No-Fire Region (Blue), from September 1 to November 30.

[Figure]

| Page 21: [1] Deleted | Microsoft Office User | 05/03/2018 3:52 PM |
|---|---|---|

| Page 31: [2] Deleted | Microsoft Office User | 08/01/2018 12:59 PM |
|---|---|---|

(

| Page 31: [3] Formatted | Microsoft Office User | 08/01/2018 1:00 PM |
|---|---|---|

Font:Bold

| Page 31: [4] Deleted | Microsoft Office User | 23/10/2017 2:33 PM |
|---|---|---|

The numbers in normal print correspond to the subset of the Maritime Continent **impacted by smoke** (**FIRE**), and **not impacted by smoke** (**NO-FIRE**) **during the October maximum in fires,** while the numbers in *(italics)* correspond to the **impacted by smoke (FIRE) throughout the entire fire season from September 3rd through November 9th**, based on MISR observations

| Page 31: [5] Formatted Table | Microsoft Office User | 23/10/2017 2:35 PM |
|---|---|---|

Formatted Table

| Page 31: [6] Formatted | Microsoft Office User | 23/10/2017 3:34 PM |
|---|---|---|

Font:Italic

| Page 31: [7] Formatted | Microsoft Office User | 23/10/2017 3:34 PM |
|---|---|---|

Font:Italic

| Page 31: [8] Formatted | Microsoft Office User | 23/10/2017 3:31 PM |
|---|---|---|

Font:Italic

| Page 31: [9] Formatted | Microsoft Office User | 23/10/2017 3:11 PM |
|---|---|---|

Font:Italic

| Page 31: [10] Formatted | Microsoft Office User | 23/10/2017 3:34 PM |
|---|---|---|

Font:Italic

| Page 31: [11] Formatted | Microsoft Office User | 23/10/2017 3:34 PM |
|---|---|---|

Font:Italic

| Page 31: [12] Formatted | Microsoft Office User | 23/10/2017 3:30 PM |
|---|---|---|

Font:Italic

| Page 31: [13] Formatted | Microsoft Office User | 23/10/2017 3:11 PM |
|---|---|---|

Font:Italic

| Page 31: [14] Formatted | Microsoft Office User | 23/10/2017 3:35 PM |
|---|---|---|

Font:Italic

| Page 31: [14] Formatted | Microsoft Office User | 23/10/2017 3:35 PM |
|---|---|---|

Font:Italic

| Page 31: [15] Formatted | Microsoft Office User | 23/10/2017 3:34 PM |
|---|---|---|

Font:Italic

| Page 31: [16] Formatted | Microsoft Office User | 23/10/2017 3:30 PM |
|---|---|---|

Font:Italic

| Page 31: [17] Formatted | Microsoft Office User | 23/10/2017 3:11 PM |
|---|---|---|

Font:Italic

| Page 31: [18] Formatted | Microsoft Office User | 23/10/2017 3:05 PM |
|---|---|---|

Font:Italic

| Page 31: [19] Formatted | Microsoft Office User | 23/10/2017 3:35 PM |
|---|---|---|

Font:Italic

| Page 31: [19] Formatted | Microsoft Office User | 23/10/2017 3:35 PM |
|---|---|---|

Font:Italic

| Page 31: [20] Formatted | Microsoft Office User | 23/10/2017 3:33 PM |
|---|---|---|

Font:Italic

| Page 31: [21] Formatted | Microsoft Office User | 23/10/2017 3:30 PM |
|---|---|---|

Font:Italic

| Page 31: [22] Formatted | Microsoft Office User | 23/10/2017 3:10 PM |
|---|---|---|

Font:Italic

| Page 31: [23] Formatted | Microsoft Office User | 23/10/2017 3:04 PM |
|---|---|---|

Font:Italic

| Page 31: [24] Formatted | Microsoft Office User | 23/10/2017 3:35 PM |
|---|---|---|

Font:Italic

| Page 31: [24] Formatted | Microsoft Office User | 23/10/2017 3:35 PM |
|---|---|---|

Font:Italic

| Page 31: [25] Formatted | Microsoft Office User | 23/10/2017 3:33 PM |
|---|---|---|

Font:Italic

| Page 31: [26] Formatted | Microsoft Office User | 23/10/2017 3:29 PM |
|---|---|---|

Font:Italic

| Page 31: [27] Formatted | Microsoft Office User | 23/10/2017 3:10 PM |

Font:Italic

| Page 31: [28] Formatted | Microsoft Office User | 23/10/2017 3:00 PM |

Font:Italic

| Page 31: [28] Formatted | Microsoft Office User | 23/10/2017 3:00 PM |

Font:Italic

| Page 31: [29] Formatted | Microsoft Office User | 23/10/2017 3:35 PM |

Font:Italic

| Page 31: [29] Formatted | Microsoft Office User | 23/10/2017 3:35 PM |

Font:Italic

| Page 31: [30] Formatted | Microsoft Office User | 23/10/2017 3:32 PM |

Font:Italic

| Page 31: [31] Formatted | Microsoft Office User | 23/10/2017 3:29 PM |

Font:Italic

| Page 31: [32] Formatted | Microsoft Office User | 23/10/2017 3:09 PM |

Font:Italic

| Page 31: [33] Formatted | Microsoft Office User | 23/10/2017 3:03 PM |

Font:Not Italic

| Page 31: [34] Formatted | Microsoft Office User | 23/10/2017 3:00 PM |

Font:Italic

| Page 31: [35] Formatted | Microsoft Office User | 23/10/2017 3:35 PM |

Font:Italic

| Page 31: [35] Formatted | Microsoft Office User | 23/10/2017 3:35 PM |

Font:Italic

| Page 31: [36] Formatted | Microsoft Office User | 23/10/2017 3:32 PM |

Font:Italic

| Page 31: [37] Formatted | Microsoft Office User | 23/10/2017 3:28 PM |

Font:Italic

| Page 31: [38] Formatted | Microsoft Office User | 23/10/2017 3:08 PM |

Font:Not Italic

| Page 31: [38] Formatted | Microsoft Office User | 23/10/2017 3:08 PM |

Font:Not Italic

| Page 31: [39] Formatted | Microsoft Office User | 23/10/2017 3:02 PM |

Font:Not Italic

| Page 31: [39] Formatted | Microsoft Office User | 23/10/2017 3:02 PM |

Font:Not Italic

| Page 31: [40] Formatted | Microsoft Office User | 27/02/2018 2:58 PM |
|---|---|---|

Font:Italic

| Page 31: [41] Formatted | Microsoft Office User | 23/10/2017 2:56 PM |
|---|---|---|

Font:Not Italic

| Page 31: [42] Formatted | Microsoft Office User | 23/10/2017 2:57 PM |
|---|---|---|

Font:Italic

| Page 31: [43] Formatted | Microsoft Office User | 23/10/2017 2:57 PM |
|---|---|---|

Font:Italic

| Page 31: [44] Formatted | Microsoft Office User | 27/02/2018 2:58 PM |
|---|---|---|

Font:Not Italic

| Page 31: [44] Formatted | Microsoft Office User | 27/02/2018 2:58 PM |
|---|---|---|

Font:Not Italic

| Page 31: [45] Formatted | Microsoft Office User | 23/10/2017 2:58 PM |
|---|---|---|

Font:Italic

| Page 31: [46] Formatted | Microsoft Office User | 23/10/2017 2:56 PM |
|---|---|---|

Font:Not Italic

| Page 31: [46] Formatted | Microsoft Office User | 23/10/2017 2:56 PM |
|---|---|---|

Font:Not Italic

| Page 31: [47] Formatted | Microsoft Office User | 23/10/2017 2:58 PM |
|---|---|---|

Font:Italic

| Page 31: [48] Formatted | Microsoft Office User | 23/10/2017 2:56 PM |
|---|---|---|

Font:Not Italic

| Page 31: [49] Formatted | Microsoft Office User | 23/10/2017 2:56 PM |
|---|---|---|

Font:Not Italic

| Page 31: [50] Formatted | Microsoft Office User | 23/10/2017 2:56 PM |
|---|---|---|

Font:Not Italic

| Page 31: [51] Formatted | Microsoft Office User | 23/10/2017 2:56 PM |
|---|---|---|

Font:Not Italic

| Page 31: [51] Formatted | Microsoft Office User | 23/10/2017 2:56 PM |
|---|---|---|

Font:Not Italic

| Page 31: [52] Formatted | Microsoft Office User | 23/10/2017 2:56 PM |
|---|---|---|

Font:Not Italic

| Page 31: [52] Formatted | Microsoft Office User | 23/10/2017 2:56 PM |
|---|---|---|

Font:Not Italic

| Page 34: [53] Formatted | Microsoft Office User | 05/03/2018 3:47 PM |
|---|---|---|

Not Highlight

| Page 34: [53] Formatted | Microsoft Office User | 05/03/2018 3:47 PM |
|---|---|---|

Not Highlight

| Page 34: [53] Formatted | Microsoft Office User | 05/03/2018 3:47 PM |

Not Highlight

| Page 34: [54] Deleted | Microsoft Office User | 04/01/2018 4:26 PM |

Monthly statistics of modeled aerosol heights

| Page 34: [54] Deleted | Microsoft Office User | 04/01/2018 4:26 PM |

Monthly statistics of modeled aerosol heights

| Page 34: [54] Deleted | Microsoft Office User | 04/01/2018 4:26 PM |

Monthly statistics of modeled aerosol heights

| Page 34: [54] Deleted | Microsoft Office User | 04/01/2018 4:26 PM |

Monthly statistics of modeled aerosol heights

| Page 34: [54] Deleted | Microsoft Office User | 04/01/2018 4:26 PM |

Monthly statistics of modeled aerosol heights

| Page 34: [54] Deleted | Microsoft Office User | 04/01/2018 4:26 PM |

Monthly statistics of modeled aerosol heights

| Page 34: [55] Formatted | Microsoft Office User | 08/01/2018 1:01 PM |

Font:Not Bold

| Page 34: [55] Formatted | Microsoft Office User | 08/01/2018 1:01 PM |

Font:Not Bold

| Page 34: [55] Formatted | Microsoft Office User | 08/01/2018 1:01 PM |

Font:Not Bold

| Page 34: [56] Formatted | Microsoft Office User | 08/01/2018 1:03 PM |

Font:Italic

| Page 34: [57] Formatted | Microsoft Office User | 08/01/2018 1:05 PM |

Font:Italic

| Page 34: [58] Formatted | Microsoft Office User | 08/01/2018 1:06 PM |

Font:Italic

| Page 34: [59] Formatted | Microsoft Office User | 08/01/2018 1:07 PM |

Font:Italic

| Page 34: [60] Formatted | Microsoft Office User | 08/01/2018 1:08 PM |

Font:Italic

| Page 34: [61] Formatted | Microsoft Office User | 08/01/2018 1:01 PM |

Font:Italic

| Page 34: [62] Formatted | Microsoft Office User | 08/01/2018 1:05 PM |

Font:Italic

| Page 34: [63] Formatted | Microsoft Office User | 08/01/2018 1:05 PM |

Font:Italic

| Page 34: [64] Formatted | Microsoft Office User | 08/01/2018 1:06 PM |

Font:Italic

| Page 34: [65] Formatted | Microsoft Office User | 08/01/2018 1:07 PM |

Font:Italic

| Page 34: [66] Formatted | Microsoft Office User | 08/01/2018 1:08 PM |

Font:Italic

| Page 34: [67] Formatted | Microsoft Office User | 08/01/2018 1:01 PM |

Font:Italic

| Page 34: [68] Formatted | Microsoft Office User | 08/01/2018 1:05 PM |

Font:Italic

| Page 34: [69] Formatted | Microsoft Office User | 08/01/2018 1:05 PM |

Font:Italic

| Page 34: [70] Formatted | Microsoft Office User | 08/01/2018 1:06 PM |

Font:Italic

| Page 34: [71] Formatted | Microsoft Office User | 08/01/2018 1:07 PM |

Font:Italic

| Page 34: [72] Formatted | Microsoft Office User | 08/01/2018 1:08 PM |

Font:Italic

| Page 34: [73] Formatted | Microsoft Office User | 08/01/2018 1:01 PM |

Font:Italic

| Page 34: [74] Formatted | Microsoft Office User | 08/01/2018 1:05 PM |

Font:Italic

| Page 34: [75] Formatted | Microsoft Office User | 08/01/2018 1:04 PM |

Font:Italic

| Page 34: [76] Formatted | Microsoft Office User | 08/01/2018 1:06 PM |

Font:Italic

| Page 34: [77] Formatted | Microsoft Office User | 08/01/2018 1:07 PM |

Font:Italic

| Page 34: [78] Formatted | Microsoft Office User | 08/01/2018 1:08 PM |

Font:Italic

| Page 34: [79] Formatted | Microsoft Office User | 08/01/2018 1:02 PM |

Font:Italic

| Page 34: [80] Formatted | Microsoft Office User | 08/01/2018 1:05 PM |

Font:Italic

| Page 34: [81] Formatted | Microsoft Office User | 08/01/2018 1:04 PM |

Font:Italic

| Page 34: [82] Formatted | Microsoft Office User | 08/01/2018 1:06 PM |

Font:Italic

| Page 34: [83] Formatted | Microsoft Office User | 08/01/2018 1:07 PM |
|---|---|---|

Font:Italic

| Page 34: [84] Formatted | Microsoft Office User | 08/01/2018 1:08 PM |
|---|---|---|

Font:Italic

| Page 34: [85] Formatted | Microsoft Office User | 08/01/2018 1:02 PM |
|---|---|---|

Font:Italic

| Page 34: [86] Formatted | Microsoft Office User | 08/01/2018 1:05 PM |
|---|---|---|

Font:Italic

| Page 34: [87] Formatted | Microsoft Office User | 08/01/2018 1:04 PM |
|---|---|---|

Font:Italic

| Page 34: [88] Formatted | Microsoft Office User | 08/01/2018 1:06 PM |
|---|---|---|

Font:Italic

| Page 34: [89] Formatted | Microsoft Office User | 08/01/2018 1:07 PM |
|---|---|---|

Font:Italic

| Page 34: [90] Formatted | Microsoft Office User | 08/01/2018 1:08 PM |
|---|---|---|

Font:Italic

| Page 34: [91] Formatted | Microsoft Office User | 08/01/2018 1:02 PM |
|---|---|---|

Font:Italic

| Page 34: [92] Formatted | Microsoft Office User | 08/01/2018 1:05 PM |
|---|---|---|

Font:Italic

| Page 34: [93] Formatted | Microsoft Office User | 08/01/2018 1:04 PM |
|---|---|---|

Font:Italic

| Page 34: [94] Formatted | Microsoft Office User | 08/01/2018 1:06 PM |
|---|---|---|

Font:Italic

| Page 34: [95] Formatted | Microsoft Office User | 08/01/2018 1:07 PM |
|---|---|---|

Font:Italic

| Page 34: [96] Formatted | Microsoft Office User | 08/01/2018 1:08 PM |
|---|---|---|

Font:Italic

| Page 34: [97] Formatted | Microsoft Office User | 08/01/2018 1:02 PM |
|---|---|---|

Font:Italic

| Page 34: [98] Formatted | Microsoft Office User | 08/01/2018 1:05 PM |
|---|---|---|

Font:Italic

| Page 34: [99] Formatted | Microsoft Office User | 08/01/2018 1:04 PM |
|---|---|---|

Font:Italic

| Page 34: [100] Formatted | Microsoft Office User | 08/01/2018 1:06 PM |
|---|---|---|

Font:Italic

| Page 34: [101] Formatted | Microsoft Office User | 08/01/2018 1:07 PM |
|---|---|---|

Font:Italic

| Page 34: [102] Formatted | Microsoft Office User | 08/01/2018 1:08 PM |

Font:Italic

| Page 34: [103] Formatted | Microsoft Office User | 08/01/2018 1:02 PM |

Font:Italic

| Page 34: [104] Formatted | Microsoft Office User | 08/01/2018 1:05 PM |

Font:Italic

| Page 34: [105] Formatted | Microsoft Office User | 08/01/2018 1:04 PM |

Font:Italic

| Page 34: [106] Formatted | Microsoft Office User | 08/01/2018 1:06 PM |

Font:Italic

| Page 34: [107] Formatted | Microsoft Office User | 08/01/2018 1:07 PM |

Font:Italic

| Page 34: [108] Formatted | Microsoft Office User | 08/01/2018 1:08 PM |

Font:Italic

| Page 34: [109] Formatted | Microsoft Office User | 08/01/2018 1:05 PM |

Font:Italic

| Page 34: [110] Formatted | Microsoft Office User | 08/01/2018 1:05 PM |

Font:Italic

| Page 34: [111] Formatted | Microsoft Office User | 08/01/2018 1:04 PM |

Font:Italic

| Page 34: [112] Formatted | Microsoft Office User | 08/01/2018 1:06 PM |

Font:Italic

| Page 34: [113] Formatted | Microsoft Office User | 08/01/2018 1:07 PM |

Font:Italic

| Page 34: [114] Formatted | Microsoft Office User | 08/01/2018 1:08 PM |

Font:Italic

| Page 34: [115] Formatted | Microsoft Office User | 08/01/2018 1:05 PM |

Font:Italic

| Page 34: [116] Formatted | Microsoft Office User | 08/01/2018 1:05 PM |

Font:Italic

| Page 34: [117] Formatted | Microsoft Office User | 08/01/2018 1:04 PM |

Font:Italic

| Page 34: [118] Formatted | Microsoft Office User | 08/01/2018 1:06 PM |

Font:Italic

| Page 34: [119] Formatted | Microsoft Office User | 08/01/2018 1:07 PM |

Font:Italic

| Page 34: [120] Formatted | Microsoft Office User | 08/01/2018 1:08 PM |
| --- | --- | --- |

Font:Italic

| Page 38: [121] Deleted | Microsoft Office User | 12/10/2017 10:51 AM |
| --- | --- | --- |

**Figure 3:** Time series of daily averaged measured AOD over the fire-constrained regions of the Maritime Continent [blue], and the non fire-constrained regions of the Maritime Continent [red], as given in **Figure 1**. Circles are computed daily mean values, while dots are computed daily standard deviation bands.

[Figure]

---

## Author Response (AR3)

**Response to Referee Number 4:**

Thank you very much for taking the time to help offer substantial grammatical and readability changes. We have taken the time to carefully review the entire manuscript and have made significant changes to the wording. We have endeavored to reduce sentence length, and make the wording tighter. In addition, we have tried to more consistently use British English spelling (such as you pointed out with homogenous, which is the USA English spelling). We really appreciate your feedback, and hope that the newer version makes it easier for the audience to access and understand.

The specific changes have been included in the markups above.

[revised manuscript text omitted]